

# Biogeochemical versus biogeophysical temperature effects of historical land-use change in CMIP6

Amali A. Amali[1], Clemens Schwingshackl[1], Akihiko Ito[2], Alina Barbu[3], Christine Delire[3],
Daniele Peano[4], David M. Lawrence[5], David Wårlind[6], Eddy Robertson[7], Edouard L. Davin[8,9,10],
Elena Shevliakova[11], Ian N. Harman[12], Nicolas Vuichard[13], Paul A. Miller[6], Peter J. Lawrence[5],
Tilo Ziehn[14], Tomohiro Hajima[15], Victor Brovkin[16,17], Yanwu Zhang[18], Vivek K. Arora[19], and
Julia Pongratz[1,16]

[1]Department of Geography, Ludwig-Maximilians-Universität München, Munich, Germany
[2]Graduate School of Life and Agricultural Sciences, The University of Tokyo, Tokyo, Japan
[3]Centre National de Recherches Météorologiques, CNRM (Université de Toulouse, CNRS, Meteo-France), 42 av. Coriolis, 31057 Toulouse, France
[4]CMCC Foundation - Euro-Mediterranean Center on Climate Change, Bologna, Italy
[5]DML: NSF National Center for Atmospheric Research, Boulder, CO, USA
[6]Department of Physical Geography and Ecosystem Science, Lund University, Lund, Sweden
[7]Met Office, Fitzroy Road, Exeter, UK
[8]Wyss Academy for Nature, University of Bern, Bern, Switzerland
[9]Climate and Environmental Physics, Physics Institute, University of Bern, Bern, Switzerland
[10]Oeschger Centre for Climate Change Research, University of Bern, Bern, Switzerland
[11]National Oceanic and Atmospheric Administration, Geophysical Fluid Dynamic Laboratory, 201 Forrestal Road, Princeton, NJ 08544, USA
[12]CSIRO Environment, Commonwealth Scientific and Industrial Research Organisation (CSIRO), Canberra, ACT, Australia
[13]Laboratoire des Sciences du Climat et de l'Environnement, LSCE-IPSL (CEA-CNRS-UVSQ), Université Paris-Saclay 91191 Gif-sur-Yvette, France
[14]CSIRO Environment, Commonwealth Scientific and Industrial Research Organisation (CSIRO), Aspendale, VIC, Australia
[15]Research Institute for Global Change, Japan Agency for Marine-Earth Science and Technology, Yokohama 236-0001, Japan
[16]Max Planck Institute for Meteorology, Hamburg, Germany
[17]Center for Earth System Research and Sustainability, Universität Hamburg, Germany
[18]CMA Earth System Modeling and Prediction Centre, China Meteorological Administration, Beijing, China
[19]Canadian Centre for Climate Modelling and Analysis, Climate Research Division, Environment and Climate Change Canada, Victora, British Columbia, Canada

**Correspondence:** Amali A. Amali (A.Amali@lmu.de)



**Abstract.** Anthropogenic land-use change (LUC) substantially impacts climate dynamics, primarily through modifications in the surface biogeophysical (BGP) and biogeochemical (BGC) fluxes, which alter the exchange of energy, water, and carbon with the atmosphere. Despite the established significance of both the BGP and BGC effects, their relative contribution to climate change remains poorly quantified. In this study, we leveraged data from an unprecedented number of Earth system models (ESMs) of the latest generation that contributed to the Land Use Model Intercomparison Project (LUMIP), under the auspices of the Coupled Model Intercomparison Project Phase 6 (CMIP6). Our analysis of BGP effects indicates a range of global annual near-surface air temperature changes across ESMs due to historical LUC, from a cooling of -0.23 °C to a warming of 0.14 °C, with a multi-model mean and spread of -0.03±0.10 °C under present-day conditions relative to the pre-industrial era. Notably, the BGP effects indicate warming at high latitudes. Still, there is a discernible cooling pattern between 30° N and 60° N, extending across large landmasses from the Great Plains of North America to the Northeast Plain of Asia. The BGC effect shows substantial land carbon losses, amounting to -122±96 GtC over the historical period, with decreased vegetation carbon pools driving the losses in nearly all analysed ESMs. Based on the transient climate response to cumulative emissions (TCRE), we estimate that LUC-induced carbon emissions result in a warming of approximately 0.20±0.14 °C, which is consistent with previous estimates. When the BGP and BGC effects are taken together, our results suggest that the net effect of LUC on historical climate change has been to warm the climate. To understand the regional drivers —and thus potential levers to alter the climate—, we show the contribution of each grid cell to LUC-induced global temperature change, as a warming contribution over the tropics and subtropics with a nuanced cooling contribution over the mid-latitudes. Our findings indicate that historically, the BGC temperature effects dominate the BGP temperature effects at the global scale. However, they also reveal substantial discrepancies across models in the magnitude, directional impact, and regional specificity of LUC impacts on global temperature and land carbon dynamics. This underscores the need for further improvement and refinement in model simulations, including the consideration and implementation of land-use data and model-specific parameterisations, to achieve more accurate and robust estimates of the climate effect of LUC.

# 1 Introduction

Land-use change and land management, hereafter referred to as land-use change (LUC), can influence the climate through (1) the alteration of physical characteristics (e.g, albedo, surface roughness, and evapotranspiration) by influencing land surface processes, such as moisture, momentum, and energy fluxes (biogeophysics; BGP) and (2) the alteration of the atmospheric composition between Earth's surface and the atmosphere, primarily through changes in atmospheric $CO_2$ concentration, which affects the planet's radiative balance (biogeochemistry; BGC). These processes culminate in altering global and regional temperatures. LUC, as a term, is often used to describe an agglomeration of many processes leading to the alteration or modification of land (land use or purpose for which humans exploit land) for the purpose or function of a particular land cover through a set of practices or strategies (land management or ways humans exploit the land) aimed at optimising the use, conservation, and stewardship of land resources (Lawrence et al., 2016; Pongratz et al., 2021). While LUC-induced land cover change is





typically clearly visible (e.g., deforestation or aff-/reforestation, A/R), land management processes including fertilisation, irrigation, pesticide application, and methods of wood harvesting (selective logging vs clear-cutting), do not alter the land cover,
yet they have recently been revealed to have substantial effects on climate as well (Erb et al., 2018; Luyssaert et al., 2014).

As evidenced by previous studies and assessments (e.g., Friedlingstein et al., 2023; Jia et al., 2019; Simmons and Matthews, 2016), emissions from LUC and their associated BGP and BGC effects constitute a significant component of anthropogenic influences on climate: LUC accounts for one-third of historical $CO_2$ emissions since pre-industrial times (Jia et al., 2019). LUC was a dominant anthropogenic forcing in the pre-industrial (Ellis, 2021; Pongratz et al., 2009) as well as the industrial
eras (Hansen et al., 1998) and remains relevant at present (Findell et al., 2017), making its consideration necessary in future climate projections (Pongratz et al., 2021; Dong et al., 2019; Brovkin et al., 2013). Understanding carbon emissions from LUC is also crucial for assessing the full impact of land-based carbon dioxide removal (CDR) solutions in climate mitigation targets (Fuhrman et al., 2023; Zickfeld et al., 2023; Matthews et al., 2022). Given that LUC patterns and their impacts are often heterogeneous, distinct from that of greenhouse gases (GHGs) (Christidis et al., 2013), a thorough understanding of these
impacts is essential to proof our understanding of observed climate change, discern regional variations, effectively map and accurately attribute the drivers of observed climate change, anticipate expected patterns, and recognise potential divergence from expected patterns. Although GHG emissions resulting from ongoing LUCs, particularly deforestation and forest degradation, have garnered considerable attention, the BGP effects of LUC remain underappreciated in policy discussions, despite their acknowledged significance (Duveiller et al., 2020). Furthermore, deliberate LUC strategies aimed at climate modification,
such as CDR initiatives, often tend to emphasise $CO_2$ reduction and overlook the BGP effects of such interventions (Jia et al., 2019).

LUC significantly affects local surface temperatures through non-radiative processes, such as changes in evapotranspiration and sensible heat exchange, as well as through radiative processes initiated through changes in surface albedo (BGP effects). Both changes in turbulent fluxes and the radiative balance of the land surface impact local climate in varying degrees de-
pending on LUC type and location. A local cooling effect might result from forest cover gains (Bright et al., 2017) or loss (Williams et al., 2021) depending on the region, thus emphasising the role of A/R in climate strategies. For example, over the Northern Hemisphere, a cooling of the Earth's climate often occurs after the conversion of forests to pasture and cropland (Lawrence et al., 2016) or after deforestation in the boreal forest region (Boysen et al., 2020; Davin and De Noblet-Ducoudré, 2010; De Noblet-Ducoudré et al., 2012) due to a reduction in available net radiation at the land surface through increased
albedo. In contrast, a reduction in evapotranspiration (ET) after deforestation in the tropics generally leads to local warming (Zhu et al., 2023; Windisch et al., 2021; Lejeune et al., 2015). On the role of surface roughness, the results from idealised deforestation experiments (e.g., Davin and De Noblet-Ducoudré, 2010; Boysen et al., 2020) affirmed that converting forests to grasslands reduces surface roughness and decreases boundary layer turbulence. This leads to lower heat and water vapour transport, causing surface warming due to greater humidity and temperature gradients. Results from their simulations (Davin
and De Noblet-Ducoudré, 2010; Boysen et al., 2020) showed that global surface warming, especially over land and in the tropics, is linked to weaker turbulent exchanges that hinder energy transfer to the atmosphere, increasing outgoing longwave



radiation. In addition to the local effects (pertaining to direct effects on the surrounding area) mentioned above, LUC can also induce non-local effects, i.e, broader influence on remote regions via atmospheric circulation changes, or advection of heat and moisture (Pongratz et al., 2021; Winckler et al., 2019a). Importantly, changes in surface characteristics can differently impact

local and non-local temperature changes (De Hertog et al., 2023; Pongratz et al., 2021). Albedo changes affecting the amount of shortwave radiation absorbed by the surface, have been shown to be important for non-local effects (Breil et al., 2024; Li et al., 2023; De Hertog et al., 2023), whereas ET and surface roughness changes dominate local effects (Pongratz et al., 2021; Duveiller et al., 2018). Although our understanding of the underlying physics has improved, estimates of BGP effects remain inconsistent across modelling studies using Earth system models (ESMs), often differing in magnitude (De Hertog et al., 2023;

Winckler et al., 2019b; Arora and Montenegro, 2011), extent (Grant et al., 2023; Santos et al., 2023; Luo et al., 2022), and direction (Devaraju et al., 2018; Pongratz et al., 2010; Pitman et al., 2009) including in regional climate models (Davin et al., 2020).

The BGC effects of LUC are often quantified as losses of carbon stored in vegetation biomass and soil. Alternatively, they are measured through the change in atmospheric $CO_2$ concentration in response to LUC emissions, which contribute to increased

radiative forcing in addition to contributions from changes in other GHG fluxes, such as methane, nitrous oxide, and emissions of aerosol precursors. The impact of LUC on the carbon cycle, notably by influencing atmospheric $CO_2$ levels, implies that LUC emissions remain a relevant flux component in global climate dynamics, accounting for about half of all LUC-related GHGs emissions (Hong et al., 2021) and 12% of total anthropogenic $CO_2$ emissions of the last 20 years (Friedlingstein et al., 2023). $CO_2$ emissions from LUC are primarily due to deforestation and conversion of natural vegetation into pasture and cropland,

alongside degradation, wood harvest, decay of related products as well as peat drainage and peat burning (Friedlingstein et al., 2022a; Pongratz et al., 2021).

Prior attempts to identify the historical effects of LUC on climate include the Land-Use and Climate, Identification of Robust Impacts (LUCID; Pitman et al., 2009; De Noblet-Ducoudré et al., 2012) project. In LUCID, Pitman et al. (2009) analysed the BGP effects of historical LUC as simulated by several ESMs and attributed inconsistencies across ESMs to their implemen-

tation of LUC, depiction of crop phenology, parameterisation of albedo, and the representation of ET across various types of land cover. Using LUCID datasets, Lejeune et al. (2017) reported higher daytime warming temperatures across regions with forest versus non-forest cover. However, they also revealed the inability of ESMs to capture the observed daytime warming and nighttime cooling effects of deforestation, indicating a need for model refinement. Among other issues, a major shortfall of LUCID was the relatively small sample size of participating ESMs, often leading to inconclusive results (Pitman et al., 2009;

De Noblet-Ducoudré et al., 2012). Although LUCID stipulated a clear protocol to implement LUC, it failed to specify the distribution of natural vegetation. As a result, this left the outcome of LUC processes, such as forest conversion (to croplands or pastures), to the discretion of models or modellers (De Noblet-Ducoudré et al., 2012). Additionally, models used varying definitions of the term "forest" and differed in which natural vegetation type was utilised for pasture expansion. The importance of selecting specific rules for modelling land cover changes, particularly with regard to their capacity to accurately reflect the

preferential historical conversion of natural, non-forested lands into pastures, was demonstrated by Reick et al. (2013). Their



results illustrated how different strategies for modelling pasture expansion—whether preferring natural grasslands or a proportional use of forests and grasslands—can significantly impact global forest coverage. Using simulations performed under the Coupled Model Intercomparison Project Phase 5 (CMIP5) project, Brovkin et al. (2013) and Boysen et al. (2014) also revealed diverse interpretations of common land-use scenarios across ESMs, especially regarding the allocation of areas for crops and pastures. They showed that the distinct representation of land-use classes across models leads to inconsistencies in the simulation of LUC across the models. Additional differences stemmed from model-specific implementations, where simulations exhibited a wide range of responses to LUC-induced changes in land carbon storage due to differences in model assumptions and accounted processes (e.g., the treatment of deforestation biomass, the simulation of fire, $CO_2$ fertilisation effect, regrowth after land abandonment, and wood harvest). Empirical evidence (e.g., Reick et al., 2013) suggests that incorporating a rule of preferentially allocating pasture on non-forest land results in a more realistic representation of forest area reduction over time, significantly affecting global carbon emissions and forest cover in specific regions, particularly in the savannas. However, such a rule was not consistent across LUCID and CMIP5 participating models. The disparity in the distribution of natural vegetation made it difficult to interpret the effects of LUC in LUCID and CMIP5 climate projections, highlighting the need for a consistent and comprehensive implementation of land-use processes across ESMs (Reick et al., 2013). The diversity in modelling approaches and adherence to simulation protocols results in varied interpretations and implementations of LUC. While both factors affect the comparability and consistency of outcomes, adherence to protocols offers room for model improvement. In contrast, the diversity in modelling approaches also reflects the inherent uncertainty in model structure and helps to mitigate an illusion of accuracy in resulting estimates.

Against this backdrops, the Land Use Model Intercomparison Project (LUMIP; Lawrence et al., 2016) evolved to provide a unique opportunity to compare the climate impact of LUC in ESMs participating in phase 6 of the Coupled Model Intercomparison Project (CMIP6; Eyring et al., 2016). To better understand the contribution and global warming mitigation potential of LUC, the LUMIP protocol includes a dataset of reconstructed LUC and model diagnostic variables. The LUMIP dataset has proven instrumental in a number of studies, including detection and attribution of LUC effects (Grant et al., 2023), contribution from different LUC types to temperature effect (Yu and Leng, 2022), localised impacts of LUC (Tang et al., 2023), LUC impacts on soil carbon (Ito et al., 2020), as well as deforestation and forestation induced climate effects (Liu et al., 2023; Luo et al., 2023; Loughran et al., 2023; Li et al., 2022a; Luo et al., 2022; Boysen et al., 2020). Furthermore, LUMIP has been used in regional (Santos et al., 2023; Singh et al., 2020) and LUC-induced global economic inequality studies (e.g., Liu et al., 2022) due to its ability to isolate impacts due to LUC.

Our study utilises the LUMIP dataset to evaluate how LUC is implemented across LUMIP models, to quantify carbon emissions and near-surface air temperature changes due to LUC, and to estimate the relative contribution of both BGP and BGC effects to historical temperature changes. By doing so, we aim to investigate potential differences in the Earth system response of different ESMs in the controlled LUMIP setup, which offers greater uniformity and comparability across the latest generation of ESMs. This controlled setup is of high relevance as analyses based on earlier attempts like the LUCID and CMIP5 projects have been limited by inconsistencies across ESMs, such as variations in the implementation of land use, and the representation





of land cover types through different plant functional types (PFTs). Beyond LUCID and CMIP5, the latest generation of models participating in LUMIP has improved by implementing more land-use processes and land management practices, such as crop irrigation, fertilisation of cropland, wood harvest, and residue management. LUMIP aimed at greater consistency across ESMs compared to LUCID and CMIP5 by also employing the latest generation of ESMs utilising an updated harmonised land-use dataset including more detail and guidance on implementation (LUH2; Hurtt et al., 2020). Additionally, the number of models

contributing to LUMIP is substantially higher; some of which have been evaluated across a broad range of objectives (e.g., Grant et al., 2023; Santos et al., 2023; Boysen et al., 2020). Consequently, LUMIP should deliver more robust estimates of BGP and BGC effects as well as make it possible to identify more comprehensively the magnitude and extent of the model spread in LUC effects. Thus, our study represents a logical next step, bridging the gap between previous studies and advancing our understanding of LUC impacts on climate through the use of LUMIP's specialised simulations.

Leveraging this progress, we analyse the response of near-surface air temperature—a model diagnostic that is sensitive to model (land and atmosphere) structure and internal parameterisation. Near-surface air temperature is indicative of anthropogenic climate change and a key quantity for climate policy—to both BGP and BGC effects of LUC across ESMs participating in LUMIP. We also analyse the changes in land carbon pools due to LUC and, using the transient climate response to cumulative emissions (TCRE; Matthews et al., 2009), we estimate the temperature response associated with carbon emissions due to LUC.

Furthermore, we investigate the contributions of different regions to global temperature change via the BGC and BGP effects, evaluating the relative importance of BGP and BGC effects in influencing near-surface air temperature. With the inclusion of more land management processes in the models used in LUMIP, our results enhance understanding of the temperature effects of LUC across state-of-the-art models, extending beyond LUCID and CMIP5 to also underline the significance of historical LUC for future projections.

## 2 METHODS

### 2.1 Simulation Setup

In this study, we utilised two CMIP6 simulations: the "*historical*" and "*hist-noLu*" experiments. The CMIP6 historical experiment (henceforth "*historical*") is described in Eyring et al. (2016). The *historical* experiment is a coupled "concentration-driven" simulation that captures the interactions between land, atmosphere, and ocean dynamics. In this simulation, external

forcings, including anthropogenic changes in the atmospheric composition (e.g., GHGs and aerosols), solar variability, and volcanic aerosols are prescribed based on observational data. This setup facilitates the evaluation of the models' capability to reproduce historical climate change, ensures the consistency of climate model forcing and model sensitivity against observational benchmarks, and serves as a foundation for formal detection and attribution studies (Grant et al., 2023; Lawrence et al., 2016). The LUMIP historical with no land-use change experiment (henceforth, "*hist-noLu*") also aligns with the CMIP6 histor-



ical concentration-driven experiment, but with a notable exception: land use and land cover remain static at their pre-industrial
levels (here, 1850) akin to the CMIP6 pre-industrial control (*piControl*) simulation (Lawrence et al., 2016). In simpler terms,
land use and land management were kept constant at their 1850 level throughout the simulation period, which resulted in no
change in the prescribed distributions of cropland, pastureland, different crop types, land management practices, and wood
harvesting among other factors. If changes in the coverage of natural vegetation occurred in the *hist-noLu* simulation, this
was due to an ESM representing dynamics in the biogeographic distribution of natural vegetation types (if represented by the
respective ESM), but not due to LUC.

The *hist-noLu* simulation is counterfactual to the *historical* simulation, as the latter includes the observed evolution of *historical*
land use and climate based on the land use harmonisation 2 dataset (LUH2; Hurtt et al., 2020). The provision of both the
*historical* and *hist-noLu* simulations is imperative to achieving the LUC separation, thus serving as the primary criterion for
selecting the ESMs used in this study. Differences in climate between the *historical* and *hist-noLu* concentration-driven setups
can be attributed exclusively to differences in the physical properties of the land surface caused by LUC (Boysen et al., 2020).
For models with a full land carbon cycle, this set-up also permits the isolation of the land $CO_2$ fluxes as they are disturbed by
LUC. Note that because the *historical* and *hist-noLu* simulations prescribe the same $CO_2$ concentration, the altered land $CO_2$
fluxes correspond directly to the widely used net LUC flux, or "land-use emissions" as the term is more colloquially called
(Friedlingstein et al., 2023). However, because environmental conditions other than atmospheric $CO_2$ concentration, such as
climate, are simulated differently in the *historical* and *hist-noLu* simulations, the BGP effects of LUC influence climate and
create BGP feedback loops, which affect plant growth and decomposition rates only in the *historical* simulation (Pongratz
et al., 2014). These effects, however, are minor compared to the impact of atmospheric $CO_2$ concentration on global carbon
fluxes and can cancel out on a global scale (Pongratz, 2009). By contrast, if two emission-driven simulations with and without
historical LUC were compared to each other, the resulting "land-use feedback" would increase the atmospheric $CO_2$ in the
simulation with LUC over and above that due to increased fossil fuel emissions (Pongratz et al., 2014; Arora and Boer, 2010).
The change in climate in these two simulations would thus be the result of both changes in land cover (the BGP effect) and the
difference in atmospheric $CO_2$ concentration (the BGC effect). The increased atmospheric $CO_2$ in the experiment with LUC
would also stimulate plant growth, thus reducing the estimate of derived LUC emissions (Pongratz et al., 2010).

We combine the estimated land-use emissions (based on the concentration-driven simulations) with model-specific TCRE
values to transform the land-use emissions into the BGC effect on climate (see Sect. 2.3.2). Our analysis of BGC effects is
restricted to carbon and does not include the effect of other non-$CO_2$ GHG fluxes. This is due to the fact that most ESMs
still lack the capability to model fluxes such as $N_2O$, $CH_4$, and other GHGs (Resplandy et al., 2024; Chang et al., 2021).
The climate effect of these fluxes is, however, included in the forcing of the concentration-driven runs for the *historical* and
*hist-noLu* simulations.





## 2.2 Model Description

Determined by model output availability, our analysis considered data from the 13 ESMs that provide data for both the *historical* and *hist-noLu* simulations: ACCESS-ESM1-5 (Ziehn et al., 2020), BCC-CSM2-MR (Li et al., 2019), CanESM5 (Swart et al., 2019), CESM2 (Danabasoglu et al., 2020), CMCC-ESM2 (Lovato et al., 2022), CNRM-ESM2-1 (Delire et al., 2020; Séférian et al., 2019), EC-Earth3-CC and EC-Earth3-Veg (Döscher et al., 2022; Hazeleger et al., 2012), GFDL-ESM4 (Dunne et al., 2020), IPSL-CM6A-LR (Boucher et al., 2020; Lurton et al., 2020), MIROC-ES2L (Hajima et al., 2020), MPI-ESM1.2-LR (Mauritsen et al., 2019), and the UKESM1-0-LL (Sellar et al., 2019). For ease of in-text reference, we hereafter refer to these models as ACCESS, BCC, CanESM5, CESM2, CMCC, CNRM, EC-Earth3-CC, EC-Earth3-Veg, GFDL, IPSL, MIROC, MPI, and UKESM, respectively. Despite some ESMs providing multiple ensemble members for both simulations, we used only one member per ESM, at a monthly timestep, to ensure equal contribution from each model. Specifically, we used the variant label r1i1p1f1 or the next lowest variant number if r1i1p1f1 was not available. For brevity, only salient features, such as the implementation of land model processes, vegetation dynamics, and land-use change processes in the ESMs pertinent to this study are outlined in Table 1. For comprehensive specifications of each model, readers are directed to the respective primary literature (see references in Table 1). The data for both the *historical* and *hist-noLu* simulations was retrieved from the Earth System Grid Federation (ESGF; https://www.esgf-data.dkrz.de/, last accessed on 14 August 2023).

Due to the model structure, clear distinctions but also commonalities exist between how ESMs treat land management. While the LUMIP protocol specifies the LUH2 dataset to be used, its implementation across ESMs still depends on the individual model architecture. For example, in the treatment of pasture, for both EC-Earth3 models, rangeland is treated as managed pasture, not allowing any shrubs or trees to grow. In UKESM, rangeland from LUH2 is not utilised and pasture PFTs are duplicates of natural grasses with no representation of management; pasture and total crop area from LUH2 are passed to the dynamic vegetation scheme and changes in these drivers result in changes in the areas of natural, crop, and pasture PFTs. In estimating land-use emissions, a handful of the ESMs (CESM2, CMCC, and MIROC) utilise the LUH distinction between primary and secondary land area, while others (ACCESS and IPSL) sum the LUH primary and secondary land fractions such that changes in primary and secondary land area fractions correspond to those of simulated ecosystem land areas (forests, grasslands, etc.). In the computation of the carbon stored in land (cLand), for a few models (BCC, CanESM5, and CESM2), land carbon pools include the contribution from carbon stored in litter (cLitter), soil (cSoil), and vegetation (cVeg) pools. For IPSL, MIROC, MPI, and the EC-Earth3 models, this also includes carbon stored in wood products (cProduct). UKESM does not simulate litter pools but it incorporates cProduct into cVeg and cSoil. In addition to the main carbon pools (cLitter, cSoil, and cVeg), the cLitter component of CESM2, CMCC, and the EC-Earth models also incorporate carbon pools from coarse woody debris, while ACCESS additionally includes a labile carbon pool (i.e., a small fraction of soil carbon that is decomposed at time scales of days). In CanESM5, some of the removed biomass is burned while the rest is distributed into cLitter and cSoil. The CESM2 model, however, distributes removed biomass between product and litter carbon pools, while the rest is released into the atmosphere. In CMCC and MIROC, removed biomass is transferred to the product carbon pools,



while across the EC-Earth3 models, fractions of aboveground biomass is transferred to surface litter, product pools, and the atmosphere. For IPSL, woody aboveground biomass is removed to three product carbon pools, each with different residence times before being released into the atmosphere. In UKESM, an approximation of aboveground carbon is removed to three product pools with varying decay rates, with woody PFTs contributing more to the slowly decaying product pools. EC-Earth3 also has two product pools with different residence times. Generally, the partitioning of biomass into these product pools across different models is determined by the type of material, such as PFT specificity, stem, and coarse roots. For instance, stems and coarse woody roots typically contribute to pools with longer residence times due to their slower decay rates, while finer materials, such as leaves and fine roots, decompose more quickly and are assigned to pools with shorter residence times. This material-specific partitioning ensures that each model can accurately simulate the carbon dynamics by accounting for the varying decomposition rates and the eventual release of carbon back into the atmosphere. In this configuration, in UKESM for example, the stem component of vegetation carbon arguably includes big roots, while the root component only represents fine roots. In the absence of explicit litter carbon pools, fine root carbon is directly added to the soil carbon pools.

In the absence of observational data, the plots and tables included in this study include reference data from the Global Carbon Budget 2023 (GCB2023; Friedlingstein et al., 2023) whenever possible. In the GCB, land-use emissions are simulated by three bookkeeping models, which are semi-empirical models that combine LUC reconstructions with information on carbon densities for different vegetation types and specific regrowth and decay curves to simulate changes in vegetation, soil, and product carbon pools. As much of this information is based on observational data or multi-model means, the GCB estimates can be seen as an independent estimate to compare to that is additionally likely closer to observations than the ESM estimates. A direct comparison to carbon pools or fluxes is not possible, since observational data comprises both natural and anthropogenic effects, such that the LUC effects alone are not separable (Pongratz et al., 2010). To contrast the change in land carbon between year 1850 and the present, we compare the spread across the LUMIP estimates of carbon stored in soil (cSoil) and vegetation (cVeg) with estimates from the "Trends and drivers of the regional-scale sources and sinks of carbon dioxide" (TRENDY v11; Sitch et al., 2015) simulations (http://sites.exeter.ac.uk/trendy/, last accessed 11 November 2023). In total, sixteen TRENDY models were used (see Supplementary Table S1). Configuration details of the TRENDY simulations can be found in Sitch et al. (2015) and Obermeier et al. (2021). Land-use change in the TRENDY models is computed by contrasting the S2 simulation (simulation without land-use change) and the S3 simulation (simulation with land-use change) of the respective models. The TRENDY-S3 simulation uses a similar land-use forcing as the LUMIP simulations, i.e. the land use harmonisation dataset (LUH2; Hurtt et al., 2020), though in updated form (Friedlingstein et al., 2023; Chini et al., 2021). Some of the TRENDY models also serve (though partly in older model versions) as land models of some ESMs used in this study.

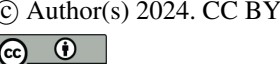



**Table 1. CMIP6 models used in this study and their implementation of land-use processes.** The columns indicate whether the model's land surface component has representations of dynamic vegetation (biogeographic shifts of vegetation types in response to environmental changes), nitrogen cycle, phosphorus cycle, subgrid-scale land-use transitions (referring to changes in land use that occur within a smaller area than the grid cell), irrigation of croplands, crop harvest, and wood harvest. NPP refers to net primary productivity and PFT refers to plant functional type.

| CMIP6 Model | Land Surface Model | spatial resolution (lat x lon) | Land model processes and vegetation dynamics | | | Land-use/land management representation | | | | | |
| --- | --- | --- | --- | --- | --- | --- | --- | --- | --- | --- | --- |
| | | | dynamic vegetation | nitrogen cycle | phosphorus cycle | PFTs | Crop PFTs | subgrid-scale land-use transitions | cropland irrigation | crop harvest | wood harvest |
| ACCESS-ESM1-5 | CABLE 2.4 | 1.25° × 1.88° | no | yes | yes | 11 | 2 | net | no | no | yes, area-based |
| BCC-CSM2-MR | BCC-AVIM 2.0 | 1.13° × 1.13° | yes | no | no | 16 | 2 | gross | no | yes, removed to the atmosphere | no |
| CanESM5 | CLASS-CTEM v1.2 | 2.81° × 2.81° | yes | no | no | 9 | 2 | net | no | yes, harvest when (if) LAI (weather) exceeds a certain threshold | no |
| CESM2 | CLM 5.0 | 0.94° × 1.25° | no | yes | no | 22 | 8 | net | yes | yes, grain to 1-yr product pool, residue to litter pools | yes, mass-based |
| CMCC-ESM2 | CLM 4.5 | 0.94° × 1.25° | no | yes | no | 16 | 1 | net | no | no | yes, LUH2 area-based |
| CNRM-ESM-1 | ISBA-CTRIP | 1.41° × 1.41° | no | no | no | 16 | 3 | net | no | no | area-based; only if forest is reduced |
| EC-Earth3-CC | HTESSEL + LPJ-GUESS 4.0 | 0.70° × 0.70° | yes | yes | no | 19 | 5*2 | gross | yes | yes, at maturity to the atmosphere | area; only if forest is reduced |
| EC-Earth3-Veg | HTESSEL + LPJ-GUESS 4.0 | 0.70° × 0.70° | yes | yes | no | 19 | 5*2 | gross | yes | yes, at maturity to the atmosphere | area; only if forest is reduced |
| GFDL-ESM4.1 | GFDL-LM 4.1 | 1.00° × 1.25° | yes | no | no | - | 2 | gross | no | yes, annually with prescribed schedules | yes, area-based |
| IPSL-CM6A-LR | ORCHIDEE v2.0 | 1.26° x 2.50° | no | no | no | 15 | 2 | net | no | yes, fixed fraction of NPP to the atmosphere | yes, mass-based |
| MIROC-ES2L | MATSIRO 6.0 + VISIT-e v1 | 2.81° × 2.81° | no | yes | no | 12 | 1 | gross | no | yes, (10% of foliage biomass) | yes, area-based |
| MPI-ESM1-2-LR | JSBACH 3.2 | 1.88° × 1.88° | yes | yes | yes | 13 | 2 | gross | no | yes, fixed fraction of litter to the atmosphere | yes, mass-based |
| UKESM1-0-LL | JULES-ES-1.0 | 1.25° × 1.88° | yes | yes | no | 13 | 2 | net | no | yes, fixed fraction of storage organ to harvest pool | no (area-based only if forest is reduced) |

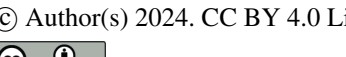




Table 1. CMIP6 models used in this study and their implementation of land-use processes (Continued). Similar to Table 1a but with additional details. The columns indicate whether the model's land surface component has representations of grazing, rangeland, pasture, tillage, nitrogen fertilisation, and shifting cultivation. PFT refers to plant functional type, nat veg refers to natural vegetation, and LAI refers to leaf area index. The changes (by area) of tree cover, crop cover, natural grassland, and pasture across the individual models are shown in Supplementary Figs. S13 - S16.

| CMIP6 Model | Land-use/land management representation | | | | | | | References |
| --- | --- | --- | --- | --- | --- | --- | --- | --- |
| | pasture | pasture harvest | grassland | rangeland | grazing | tillage | nitrogen fertilisation | |
| ACCESS-ESM1-5 | replace nat veg by a grassland/pasture PFT | no | yes | no | no | no | no | Ziehn et al. (2020), Kowalczyk et al. (2013) |
| BCC-CSM2-MR | no | no | no | treated as nat veg | yes | no | no | Wu et al. (2019), Li et al. (2019) |
| CanESM5 | no | no | yes | - | no | yes | no | Swart et al. (2019), Melton et al. (2020) |
| CESM2 | no, C3 and C4 grass | no | yes | treated as nat veg | no | no | yes | Danabasoglu et al. (2020), Lawrence et al. (2019) |
| CMCC-ESM2 | no, C3 and C4 grass | no | yes | treated as nat veg | no | no | yes | Lovato et al. (2022), Oleson et al. (2013) |
| CNRM-ESM-1 | no, C3 and C4 grass | no | yes | treated as nat grass | no | no | no | Séférian et al. (2019), Delire et al. (2020) |
| EC-Earth3-CC | C3 and C4 grass replace nat veg by a grassland/pasture PFT | yes (1x /yr) | yes | added to pasture | yes | yes | yes | Döscher et al. (2022), Balsamo et al. (2009), Smith et al. (2014) |
| EC-Earth3-Veg | C3 and C4 grass replace nat veg by a grassland/pasture PFT | yes (1x /yr) | yes | added to pasture | yes | yes | yes | Döscher et al. (2022), Balsamo et al. (2009), Smith et al. (2014) |
| GFDL-ESM4.1 | yes | daily grazing to minimum LAI | yes | yes | yes | no | yes | Dunne et al. (2020), Shevliakova et al. (2024) |
| IPSL-CM6A-LR | no, C3 and C4 grass | no (possible offline) | yes | treated as nat grass | no | no, turnover could be changed | no (possible in new version) | Boucher et al. (2020), Lurton et al. (2020) |
| MIROC-ES2L | grazed nat veg | represented as increased mortality of foliage | no | yes, no deforestation when converting to rangeland | yes, on pasture and rangeland | - | - | Hajima et al. (2020), Ito and Hajima (2020) |
| MPI-ESM1-2-LR | no, C3 and C4 grass | no harvest but grazing rate | yes | treated as nat veg or pasture | yes | no | no | Mauritsen et al. (2019), Reick et al. (2013) |
| UKESM1-0-LL | C3 and C4 grass replace nat veg by a grassland/pasture PFT | no | yes | treated as nat veg | no | no | yes | Sellar et al. (2020), Wiltshire et al. (2020) |





## 2.3 Data Analysis and Statistical Methods

### 2.3.1 Isolating land-use change effects

The difference in near-surface air temperature between the *historical* and *hist-noLu* simulations is used here to isolate the BGP effect on temperature attributable to LUC. For each model, we express the change in near-surface air temperature as:

$$\Delta T_{bgp}(n,x,t) = T_{historical}(n,x,t) - T_{hist-noLu}(n,x,t) \tag{1}$$

where $T$ is the near-surface air temperature (CMIP6 variable: tas), $\Delta T_{bgp}$ is the effect due to LUC as a function of time ($t$), in the grid cell ($x$), and the index $n$ indicates the $n$-th model. The equation is valid for the quantification of the global mean
response as well as for the temperature change from LUC at any given grid cell. Similarly, the contrast in carbon pools between the *historical* and *hist-noLu* simulations yields the BGC effect on carbon attributable to LUC:

$$\Delta C(n,x,t) = C_{historical}(n,x,t) - C_{hist-noLu}(n,x,t) \tag{2}$$

where $\Delta C$ is the change in any of the carbon pools: cLand, cLitter, cSoil, and cVeg due to LUC. We used the cLand variable for our analysis of LUC and where not available, cLand was computed as a summation of cLitter, cSoil, and cVeg. For IPSL,
this also includes cProduct.

All spatial representations in this study depict the mean over the last three decades of the *historical* and *hist-noLu* simulations (spanning from 1985 to 2014) for both climate and carbon metrics, whereas temporal plots are presented as a 10-year running mean. To distinguish the BGP effects from internal climate variability, we employ the modified Student's t-test adjusted for spatial auto-correlation (Lorenz et al., 2016; Zwiers and Von Storch, 1995) to identify grid cells with statistically significant
changes at the 5% significance level.

### 2.3.2 Global temperature response and local contributions to global temperature change

To quantify the temperature response from the change in land carbon stocks we use an approximation by the "transient climate response to cumulative carbon emissions" (TCRE; see Gillett et al., 2013; Matthews et al., 2009). TCRE is expressed as the ratio of the transient climate response (TCR) to cumulative fossil fuel emissions (Leduc et al., 2016; Matthews et al., 2009),
where TCR is defined as the global temperature change at the time of $CO_2$ doubling in a simulation with a 1% per year compounded $CO_2$ increase (*1pctCO2* simulation in CMIP6). TCR is computed as the change in the global average surface temperature over 20 years, centred at $CO_2$ doubling (years 60 to 79 in the *1pctCO2* simulation), relative to the same period in the pre-industrial control simulation and smoothed with a 140-year linear fit to correct for residual drift (Meehl et al., 2020). While TCR focuses on the radiative response of temperature to increased atmospheric $CO_2$ concentrations, TCRE
additionally considers the dynamics of land and ocean carbon sinks, which influence the amount of cumulative fossil fuel



emissions necessary to double atmospheric $CO_2$ concentrations. TCRE identifies the amount of global warming ($\Delta T$) per unit cumulative fossil fuel emission at the time of atmospheric $CO_2$ concentrations doubling relative to the pre-industrial baseline in the *1pctCO2* simulation, expressed as °C EgC$^{-1}$ (where 1 exagram of carbon=$10^{18}$gC). Empirical studies (e.g., Leduc et al., 2016, 2015; Gillett et al., 2013; Matthews et al., 2009) have consistently shown that the TCRE is approximately constant over time and independent of the emission trajectory, underscoring a near-linear relationship between cumulative $CO_2$ emissions and global temperature change. Furthermore, the spatial pattern of temperature change per degree of global warming has been shown to remain approximately constant with increasing global mean temperature (Gillett et al., 2013). In prior research, Arora et al. (2020) identified TCRE ratios for an array of CMIP6 models, from which we retrieve the TCRE value for each model used in this study, adding the TCRE for CMCC-ESM2 derived by Lovato et al. (2022). For the two EC-Earth3 models, no TCRE values are available from the literature, and we thus exclude them from the TCRE analysis. As applied across earlier studies (e.g., Kondo et al., 2022; Boysen et al., 2020; Leduc et al., 2016; Boysen et al., 2014), we integrated the derived TCRE metrics with changes in land carbon fluxes ($\Delta cLand$) to estimate the temperature response to the change in land carbon fluxes. For each model, we express the global mean temperature response as

$$\Delta \overline{T}_{bgc}(n,t) = -TCRE(n) * \Delta \overline{c}Land(n,t) \tag{3}$$

where $\Delta T_{bgc}$ is the global near-surface air temperature change due to BGC effects of LUC, $\Delta \overline{c}Land$ is the diagnosed value of each model's global cumulative $CO_2$ emissions from LUC, calculated as the total change in the land carbon content between the *historical* and *hist-noLu* simulations, and the TCRE value is obtained for each model from Arora et al. (2020) and Lovato et al. (2022). The overline indicates globally averaged values. The minus sign accounts for the fact that a decrease in land carbon content ($\Delta C$, estimated from Equation (2) on the grid cell level, then summed globally) corresponds to increasing atmospheric carbon content and thus a temperature increase.

Given that the regional patterns of temperature change scale approximately linearly to the cumulative $CO_2$ emissions (Leduc et al., 2016, 2015; Matthews et al., 2009), we use this to create a spatial pattern associated with the global mean temperature change due to the BGC effects (Equation (3)). For this purpose, we utilised a simple linear regression to obtain the regional-to-global ratio of temperature (RGRT) for each model, a process also known as 'simple pattern scaling' (Mitchell, 2003; Tebaldi and Arblaster, 2014);

$$T(n,x,t) = a(n,x) * \overline{T}_{glob}(n,t) \tag{4}$$

Here, the slope $a$ represents the RGRT and $\overline{T}_{glob}$ represents the global mean temperature (GMT) for each model. We use data from the *1pctCO2* simulation for a period ranging between 150 to 165, depending on the model, to estimate both the GMT and the grid cell temperature with which the slope was derived. The estimated grid cell slope, $a(n,x)$ is hereafter combined with $\Delta \overline{T}_{bgc}(n,t)$, to quantify the temperature response to the BGC effects for each model and each grid cell over time:

$$\Delta T_{bgc}(n,x,t) = a(n,x) * \Delta \overline{T}_{bgc}(n,t) \tag{5}$$





We further attempt to distinguish between the grid cell temperature *contribution* and the grid cell temperature *effect* - the two different underlying questions are how much an individual location (grid cell) of LUC contributed to the global signal versus how the climate in a specific location (grid cell) is affected by global LUC. We quantify the grid cell contribution to the global

signal by estimating how carbon emissions in each grid cell due to historical LUC contribute to the estimated global BGC-induced temperature change. For this purpose, we multiply the change in land carbon due to LUC for each model and in each grid cell with each model's TCRE value:

$$\Delta T_{\text{bgc}}^{\text{local}}(n,x,t) = -\Delta C(n,x,t) * TCRE(n) \tag{6}$$

$\Delta T_{\text{bgc}}^{\text{local}}$ differs from $\Delta T_{bgc}$ estimated in Equation (5), as the former quantifies the contribution of carbon emissions due to LUC

in each grid cell to the global temperature change, while the latter quantifies the local temperature change caused by the total, global carbon emissions due to LUC.

Finally, to estimate how the BGP effects in each grid cell contribute to the global BGP-induced temperature change, we multiply the grid cell estimated temperature change, $\Delta T_{bgp}(n,x,t)$, with the grid cell weighted area, where the grid cell weighted area is expressed as the ratio of the grid cell area, $A_{grid}$ for each model ($n$), to the Earth's surface area ($A_{SFC}$). We express each

grid cell's BGP contribution as:

$$\Delta T_{\text{bgp}}^{\text{local}}(n,x,t) = \Delta T_{bgp}(n,x,t) * \frac{A_{grid}(n,x)}{A_{SFC}} \tag{7}$$

### 2.3.3 Descriptive Statistics

We applied Equations (1) through (7) to the spatial fields of each CMIP6 model (differentiating between grid cell and global metrics) and subsequently computed the ensemble statistics for 13 models and 11 models for the BGP and BGC estimates,

respectively. We excluded EC-Earth3-CC and EC-Earth3-Veg from the analysis of the BGC effects as the ocean component needed to estimate the TCRE value is missing in the former, while the latter has no fully activated C-cycle. For the BGP analysis, spatial maps and estimates are computed as the mean of 1985-2014, while for the BGC analysis, we used the value at the end of the simulation period (year 2014), which represents the cumulative emissions from 1850-2014. Evidenced by previous studies (e.g., Hajima et al., 2024; Séférian et al., 2020; Gier et al., 2020; Collier et al., 2018), we interpret the multi-

model mean across ESMs as the most accurate representation of the global estimates, while the standard deviation across the model estimates delineates the associated inter-model uncertainty. The signal-to-noise ratio was computed by dividing the multi-model mean by the standard deviation across the models, whereas the model agreement was computed by summing the direction of change (+1 or -1) of individual grid cells for each model. For spatial representations, we interpolated the results of each model using the Climate Data Operator (CDO; Schulzweida, 2023) onto a uniform grid, using a spatial resolution already

common to some of the ESMs: 0.94° x 1.25° (latitude x longitude). For extensive variables, such as land-use emissions, we used conservative remapping with the 'remapcon' function to preserve the integrals of the global totals. For intensive variables, such as temperature, we used bilinear interpolation with the 'remapbil' function to preserve the mean values.





# 3 RESULTS

## 3.1 Contribution of carbon pools to cumulative land emissions

The global multi-model mean carbon loss due to historical LUC is -122±96 GtC cumulatively over the period 1850-2014 (Table 2). The upper bound aligns with reference values in GCB2023 (Friedlingstein et al., 2023), providing a useful comparison. But the spread in the magnitude of LUC emissions among the ESMs is immediately apparent, with five ESMs—MIROC, CMCC, GFDL, CESM2, and EC-Earth3-Veg—yielding estimates very close to those in GCB2023 (Fig. 1a). Out of the considered ESMs, the EC-Earth3-CC model simulated the largest historical decrease in total land carbon with up to 314 GtC

in carbon loss, whereas CNRM shows the smallest decrease in land carbon of about 3 GtC. By contrast, BCC is the only model simulating a gain in land carbon due to historical LUC of about 26 GtC. Additionally, the trajectory of change in total land carbon, $\Delta$cLand (Fig. 1a), highlights variations in how ESMs simulate changes in land $CO_2$ fluxes, often manifesting as distinguishable model clusters. Notably, CMCC, IPSL, and UKESM form a distinct cluster. These models share a common approach by implementing net sub-grid transitions and simulating grasslands, but they do not represent pasture or grazing.

This specific model characteristic likely leads to similar trajectories in land $CO_2$ fluxes. Boysen et al. (2020) also suggests that such model configurations can significantly influence land carbon dynamics, thereby explaining the observed model clustering. The GCB2023 multi-model decadal estimates of land carbon emissions from the 1960s to 2000s, included in Fig. 1c, show that the multi-model mean estimate lies substantially outside the uncertainty range of the decadal mean estimates; only the ESMs with a high rate of LUC emissions (approximately half of the ESMs) fall inside the GCB2023 uncertainty range. While

the long-term (cumulative) emissions from LUC are captured reasonably well by the ESMs as shown in Fig. 1a, the annual and decadal emissions estimates in Fig. 1c aligns more closely with the GCB2023 estimates. This closer alignment is due to annual (decadal) estimates being more responsive to recent changes in land use practices, policies, and socio-economic conditions, such as deforestation and agricultural expansion. In contrast, long-term (cumulative) estimates smooth out year-to-year variations, which can obscure recent trends and compound discrepancies over time.

The increase in $\Delta$cLand in BCC (Fig. 1a) results from a carbon gain due to LUC and can be partly traced to increasing carbon content in its litter and soil carbon pools (Figs. S1b-c, and S3 - S5), whereas vegetation carbon ($\Delta$cVeg) shows an almost steady decrease - in line with (though smaller than) the other models (Figs.S1a and S2). Boysen et al. (2020) reported an increasing trend in $\Delta$cVeg for BCC over the temperate regions outside deforested areas, where cooling and precipitation increases overlap, leading to a higher gross primary productivity. This disparity may suggest that while BCC shows a decrease in $\Delta$cVeg globally,

specific regional increases in temperate areas could indicate a complex interaction between land carbon pool treatments in BCC and regional climate dynamics over the Northern Hemisphere. Additionally, carbon transfer in BCC from deforested carbon to soil carbon pools instead of the atmosphere could account for the simulated increase. Similar to BCC, the small $\Delta$cLand change also in CNRM arises from contrasting changes in its contributing pools: specifically, a decrease in $\Delta$cVeg contrasts with increases in both the litter and soil carbon pools. In CNRM, grasses, which have a higher root-to-shoot ratio, contribute





**Table 2. Changes in global mean near-surface air temperature (T, in °C) and global ΔcLand (sum of ΔcSoil, ΔcVeg, and ΔcLitter) due to biogeophysical (bgp) and biogeochemical (bgc) impacts of land-use change for the 13 Earth System Models (ESMs) considered in this study.** Values in parentheses denote the standard deviation, estimated as the global spread across each ESM, and as the global model spread across the set of ESMs in the multi-model mean. The standard deviation for $\Delta T_{bgc}$ is less than 0.01 for all the models and therefore not included. TCRE values for CMCC-ESM2 are obtained from Lovato et al. (2022) and for all other models from Arora et al. (2020). The multi-model mean and standard deviation is computed across the set of ESMs. The model marked * (EC-Earth3-Veg*) is excluded from the multi-model mean of ΔcLand because it has no fully activated carbon cycle. For the two EC-Earth3 models, no TCRE values are available from the literature hence $\Delta T_{bgc}$ for these models were not computed.

| Models | $\Delta \mathbf{T}_{bgp}$(°C) | $\Delta$ **cLand** (GtC) | $\Delta \mathbf{T}_{bgc}$ (°C) | TCRE (°C EgC$^{-1}$) |
|---|---|---|---|---|
| ACCESS-ESM1-5 | 0.14 (0.17) | -39 (4) | 0.08 | 2.02 |
| BCC-CSM2-MR | -0.23 (0.13) | 26 (2) | -0.03 | 1.32 |
| CanESM5 | -0.07 (0.14) | -135 (12) | 0.28 | 2.09 |
| CESM2 | -0.02 (0.16) | -186 (8) | 0.40 | 2.13 |
| CMCC-ESM2 | -0.07 (0.17) | -101 (5) | 0.21 | 2.08* |
| CNRM-ESM-1 | -0.01 (0.13) | -3 (3) | 0.01 | 1.63 |
| EC-Earth3-CC | 0.13 (0.17) | -314 (18) | - | - |
| EC-Earth3-Veg* | -0.01 (0.16) | -243 (12) | - | - |
| GFDL-ESM4 | -0.17 (0.10) | -182 (12) | 0.26 | 1.45 |
| IPSL-CM6A-LR | 0.04 (0.22) | -58 (4) | 0.12 | 2.13 |
| MIROC-ES2L | -0.02 (0.23) | -157 (7) | 0.22 | 1.39 |
| MPI-ESM1-2-LR | 0.00 (0.13) | -198 (17) | 0.33 | 1.65 |
| UKESM1-0-LL | -0.08 (0.14) | -125 (10) | 0.29 | 2.3 |
| Multi-model mean (Std. dev.) | -0.03 (±0.10) | -122 (±96) | 0.20 (±0.15) | |

more below-ground litter fall than trees, leading to accumulation in soil carbon pools (Boysen et al., 2020, 2021). This is in addition to crop harvest not being represented in CNRM, which could lead to overestimation in its cLitter. For all models except BCC and CanESM5, LUC-driven changes in the total land carbon are primarily caused by changes in cVeg across most grid cells (Figs. 1a and S1). In contrast, for BCC and CanESM5, changes in cLand are predominantly influenced by changes in cSoil, with both models simulating an increase in cSoil due to LUC.

We now focus on the spatial patterns of ΔcLand and provide additional details of the different carbon pools contributing to the total land carbon in the Supplements. Collectively, the ESMs spatially depict a widespread depletion in cLand (Fig. 2a), a pattern that is more obvious over certain regions. Specifically, there is substantial and spatially coherent depletion in ΔcLand across regions spanning western to eastern Africa, the eastern U.S., southern Brazil, and southeast Asia (Fig. 2a).





The inter-model variability among ESMs (Fig. 2b) reveals a pattern: regions with noticeable inter-model variability correspond
to regions registering peak losses in cLand. There is a high signal-to-noise ratio (Fig. 2c) over regions with large change in
cLand, albeit with magnitude variations across ESMs (Fig. S2). This is corroborated by the inter-model agreement (Fig. 2d),
which shows general agreement in the direction of change of land $CO_2$ fluxes in the regions that exhibit large changes in
land carbon content. While overarching commonalities may exist across ESMs in regions impacted by LUC, disparities exist
in the specifics, distribution, and intensities. This reflects the complexity of LUC impacts, which can both sequester (e.g.,
A/R) or release (e.g., deforestation) carbon. For example, CNRM and BCC show pronounced increases in cLand over several
regions, whereas models like ACCESS and CMCC show more muted changes (Fig.S2) with the muted changes in ACCESS
likely due to low representation of land management practices among other reasons. The EC-Earth3 models show a loss in
cLand, which is stronger than the other models evaluated. Over the polar regions, MIROC and IPSL show clear changes, while
others like CMCC and UKESM have no noticeable changes. Over Africa and Australia, responses also vary among models,
with models like MIROC and MPI depicting more obvious changes compared to ACCESS, BCC, and CNRM. Some ESMs,
including EC-Earth3-CC, EC-Earth3-Veg, GFDL, and UKESM, reveal obvious carbon pool reduction over Siberia (Fig. S2), a
signal ambiguous across other models. Additionally, while both BCC and CNRM distinctly simulate an increase in land carbon
storage over North America - a trend also mirrored in their simulation of ΔcSoil - BCC is the only model indicating an increase
in ΔcLand in that region (Figs. 1a, c, and S2).

Furthermore, while the decrease in ΔcLand for CanESM5 is attributable to the decrease in ΔcVeg and ΔcSoil, similar to
most models (while ΔcLitter shows an increase), the decrease in ΔcSoil is stronger and much steeper beyond 1900 relative
to the other models. Except for BCC and CNRM, tropical changes dominate the decline in land carbon; a change mirrored
in distribution (mid-latitudes) and direction (increase) in both BCC and CNRM (Figs. S2 and S3). In models like CESM2
and GFDL, a decrease in the ΔcLand is already visible at the start of the simulation (not shown); a decrease disproportionate
across other models and we attribute this to models' treatment of pre-1850 land use. Notwithstanding these differences, our
estimates of ΔcLand fall within the range simulated across dynamic global vegetation models (DGVMs) (grey shading in Fig.
S1), capturing most individual model estimates. In exploring these differences, we note that while a handful of models indicate
ΔcLand estimates to be equal to the sum of changes in the cVeg, cSoil, and cLitter pools, this is not consistent across all
the models, giving rise to what we prefer to term "residuals" (Fig. S12). For some of the ESMs, this residual is equal to the
carbon stored in the product pools (cProduct; see Fig. S1c), while for others it is non-existent, reflecting how different models
implement LUC.

## 3.2 Biogeochemical effects of land-use change

We hereafter estimate the temperature response to cumulative LUC emissions across the ESMs to determine the effect of
resulting LUC emissions on the climate (Table 2). Our estimates of the overall global mean temperature response ($\Delta T_{bgc}$) to
historical LUC land $CO_2$ fluxes demonstrate considerable variation across models. The globally-averaged mean temperature

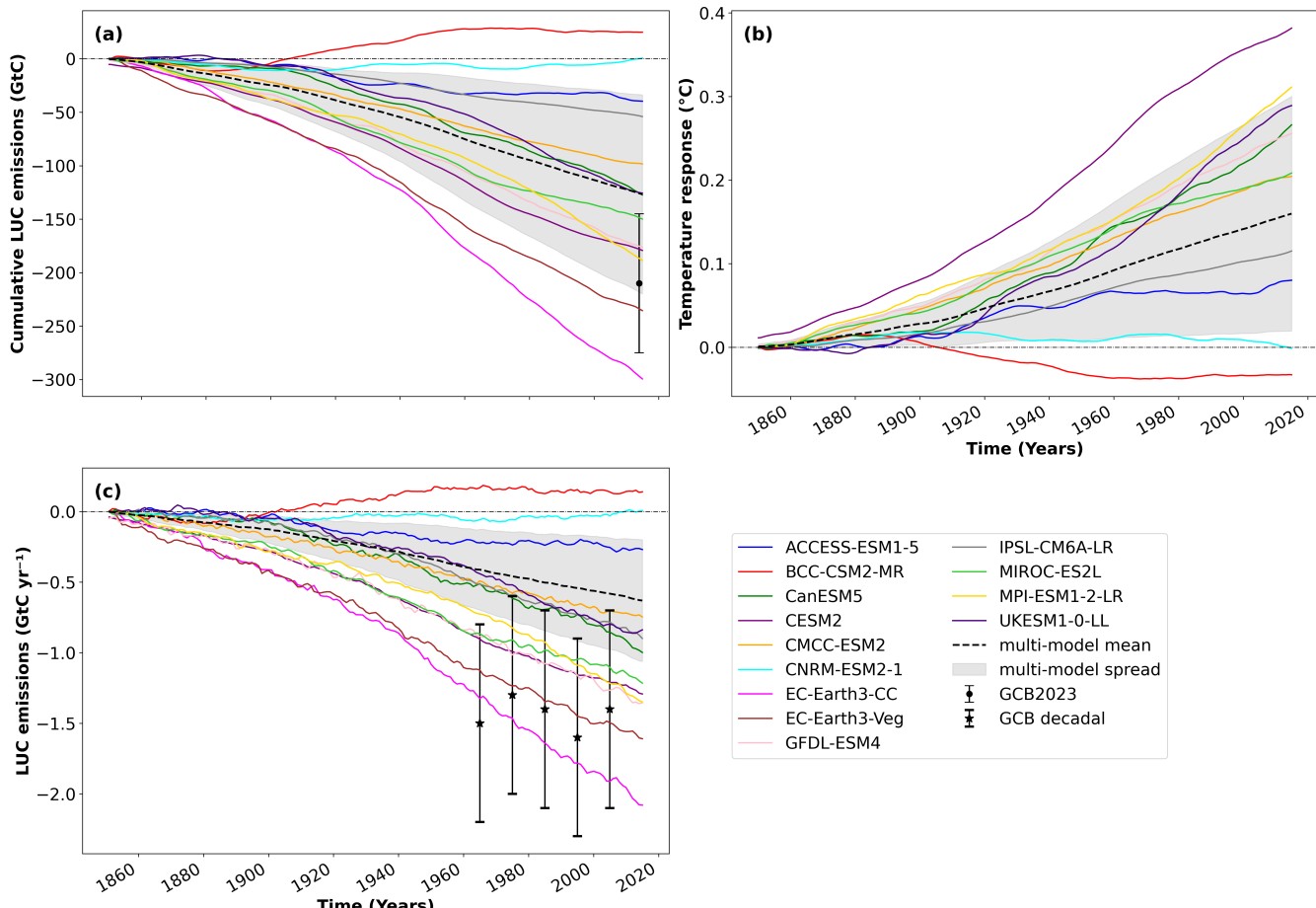

**Figure 1.** Time series of change in (**a**) total land carbon pools (ΔcLand) (**b**) global temperature response to cumulative carbon emissions and (**c**) LUC emissions (GtC yr⁻¹) due to biogeochemical effects of land-use change (LUC) as simulated by CMIP6 Earth system models (ESMs). A 10-year running average is applied. The black dot with whiskers in panel (**a**) represents the mean and standard deviation in ΔcLand estimates of the Global Carbon Budget (Table 8; Friedlingstein et al., 2023), which is based on simulations from three bookkeeping models with uncertainties quantified using dynamic global vegetation models (DGVMs). In panel (**c**), data-based estimates of decadal mean net LUC emissions for the 1960s, 1970s, 1980s, 1990s, and 2000s from the Global Carbon Budget are overlaid as an asterisk (*) with uncertainty ranges from Table 7 of Friedlingstein et al. (2023). The thick dotted black line and the grey shaded area represent the multi-model mean estimate and the standard deviation across 13 ESM estimates for (**a**) and (**c**) and 11 ESM estimates for (**b**)



change (Table 2) ranges from cooling of -0.03°C (BCC) to a warming of 0.40°C (CESM2), with a multi-model mean (standard deviation) of 0.20 (±0.14) °C (Fig. 3, Table 2). The spatial patterns of the multi-model mean of $\Delta T_{bgc}$ shows warming throughout the globe (Fig. 3a) with a clear Arctic amplification, as expected as response to a GHG forcing (Fig. 3a) and consistent with previous findings on impact of GHG (e.g., Rantanen et al., 2022; Kornhuber and Tamarin-Brodsky, 2021;

Cohen et al., 2018). While the ESMs generally agree on the direction of the BGC-induced temperature change (Fig. 3d), the spread in magnitude (Fig. 3b) suggests considerable inter-model variability over the high latitudes relative to the mid-and low latitudes and over land, a pattern similar to that observed in the multi-model mean. We already see this spread in the evolution of models' responses to the cumulative land $CO_2$ fluxes (Fig. 1b), with a gradual but consistent trend of increasing warmth since the pre-industrial era. We observe a wider dispersion across the ESMs' temperature response at the end of the

simulation (year 2014); a spread that also highlights the divergence and variability across models' TCRE estimates (Lamboll et al., 2023; Canadell et al., 2021; Matthews et al., 2009). BCC evolves like the other ESMs but begins a gradual descent post-1900, making it distinct from other ESMs by being the sole model to simulate an overall cooling in response to gain in the land carbon due to the historical LUC mentioned earlier. A similar but delayed decline (towards zero) is observed in CNRM after a prolonged period of relatively stable $\Delta T_{bgc}$, making it the only model that agrees with BCC. In ACCESS, BCC, and

CNRM, the temperature response to land $CO_2$ fluxes (Fig. 1b) evolves in similar fashion to the land $CO_2$ fluxes (Fig. 1a). We note that within the overall increasing temperature trend, the CESM2 model stands out for its particularly steep increase in comparison to other models due to its relatively high TCRE value (highest after UKESM) compared to other ESMs used in this study. However, unlike CNRM, the CESM2 model does not show any distinctive behaviour either temporally or spatially.

### 3.3 Biogeophysical impacts of land-use change

We further analyse the biogeophysical effects of LUC on a global scale (Fig. 4) as the multi-model mean of near-surface air temperature ($\Delta T_{bgp}$) for a 30-year timeframe (1985–2014). Our results demonstrate a weak global signal, ranging across models from -0.23 °C (cooling, BCC) to 0.14 °C (warming, ACCESS) with a multi-model mean (standard deviation) of -0.03 (±0.10) °C (Fig. 5, Table 2). Locally, $\Delta T_{bgp}$ remains small in many regions (Fig. 4), and the robust features (also with a high model agreement, Fig. 4d and S7) are only found in isolated regions, including a warming pattern in the North Atlantic

and a cooling over the Great Plains of the U.S. There is a tendency towards a cooling effect in the mid-to-high latitudes with a cooling strip between latitudes 30° N and 60° N, extending as a cooling band over land through eastern Europe to the Northeast Plain of Asia. We also note the more subdued warming in some tropical regions, west, and southern Africa with mixed or nuanced signals. The Arctic warming stands out especially in the EC-Earth3 and IPSL models, despite some models like BCC, CanESM, and GFDL showing a cooling pattern, and others presenting a patchwork of cooling and warming effects

(Fig. S7). Furthermore, in Fig. 4b we highlight the variability in ESM estimates, most notably in the polar regions, confirming the complexity of attributing specific patterns to the BGP effects of LUC. High model agreement is observed in the areas that exhibit the strongest temperature responses, particularly in North America, parts of Eurasia, and the North Atlantic (Fig. 4d). This agreement is nuanced by the signal-to-noise ratio (SNR), which is particularly high over North America and the North





**Figure 2.** Change in total land carbon pools (ΔcLand) as (**a**) the multi-model mean, (**b**) the inter-model spread, (**c**) the signal-to-noise ratio, and (**d**) the inter-model agreement due to biogeochemical effects of land-use change. Results were computed from 13 Earth system models as the cumulative value at the end of the simulation (year 2014). The signal-to-noise ratio (**c**) indicates the strength of the signal as compared to the inter-model uncertainty. It measures the relative weight of the multi-model mean anomalies in (**a**) with respect to the model coherence in (**b**) where a high absolute number means a robust signal. The inter-model agreement on the other hand shows the direction, rather than magnitude, of change for each grid cell (browns: negative/decreasing; greens: positive/increasing).





**Figure 3.** Estimated temperature response to cumulative LUC emissions as $(\Delta T_{bgc})$ **(a)** the multi-model mean, and **(b)** the inter-model spread, computed as the standard deviation, showing the uncertainty in estimates over each grid cell. The signal-to-noise ratio **(c)** indicates the strength of the signal as compared to the inter-model uncertainty. It measures the relative weight of the multi-model mean anomalies in **(a)** with respect to the model coherence in **(b)** where a high absolute number means a robust signal. And finally, **(d)** the inter-model agreement shows the sum of the sign of $\Delta T_{bgc}$ (-1 or +1) across all models (direction, rather than magnitude) for each grid cell (blues: negative/decreasing; reds: positive/increasing). Results computed across 11 Earth system models, as the temperature response due to the cumulative land $CO_2$ fluxes at the end of the simulation (year 2014) for each model from the difference between the *historical* and *hist-noLu* simulation.





**Figure 4.** Response of near-surface air temperature due to biogeophysical effects of LUC ($\Delta T_{bgp}$) across 13 Earth System Models (ESMs): (**a**) the multi-model mean, indicating the average temperature response (stippling indicates regions not statistically significant at the 5% level; the dashed boxes show the spatial extents of the regions considered in Fig. 5) (**b**) the inter-model spread, computed as the standard deviation across models, shows the uncertainty in estimates over each grid cell. The signal-to-noise ratio (**c**) indicates the strength of the signal as compared to the inter-model uncertainty. It measures the relative weight of the multi-model mean anomalies in (**a**) with respect to the model coherence in (**b**) where a high absolute number means a robust signal. And finally, (**d**) the inter-model agreement shows the sum of the sign of $\Delta T_{bgp}$ (-1 or +1) across all models (direction, rather than magnitude) for each grid cell (blues: negative/decreasing; reds: positive/increasing). Results are computed as the difference between the *historical* and *hist-noLu* simulations in 1985–2014.

Atlantic, indicating a rather clear BGP signal due to LUC in these regions (Fig. 4c). Conversely, the SNR is low in the higher latitudes, suggesting more uncertain estimates over these regions.

Looking into the individual ESM outputs (Fig. S7), particularly over the tropics, most ESMs show detectable changes in $\Delta T_{bgp}$, as seen over the Amazon, West and Central Africa; a change that is consistent with expectations given the extensive



LUC in these areas, particularly deforestation. Additionally, the inter-model variability becomes evident. Notably, the Northern Atlantic east of Greenland shows substantial differences among models, with several ESMs indicating a clear yet opposite
signal of $\Delta T_{bgp}$. Such contrast suggests that the signal in Fig. 4a could be more by chance of the large-warming models being one more than the large-cooling models, rather than a definitive effect of LUC. In Southern Brazil, only CESM2 reveals a clear warming pattern, whereas the other models exhibit a mixed response, underlining the variation in the influence of LUC on regional climates across models. This demonstrates that while certain areas show large absolute values in temperature change due to BGP effects, the robustness of the multi-model mean is low as the signals vary significantly across ESMs, necessitating
careful consideration of model spread and underlying factors contributing to the disparity in estimates.

The evolution of the global $\Delta T_{bgp}$ due to LUC (Fig. 5a) shows a wide spread across ESMs, which slightly widens over time. However, the trends (warming and cooling) remain inconsistent across models; a trend still present when analysed across models with multiple ensemble members (not shown). Globally, we observe a smaller magnitude in $\Delta T_{bgp}$ (compared to the regional trends, Figs. 5b-g) with the multi-model mean indicating a small cooling effect. For a few regions (selected due to
their distinct cooling/warming signals, in the direction of $\Delta T_{bgp}$, (see Fig. 4a), however, the trends in $\Delta T_{bgp}$ show higher magnitudes with smaller disparity across ESMs. Furthermore, we observe higher variability across the high latitudes (Figs. 5b-d) compared to regions over the tropics (Figs. 5e-g), with a change in $\Delta T_{bgp}$ that tends towards zero from the high latitudes to the tropics.

### 3.4   Local biogeochemical versus biogeophysical response to land-use emissions

To analyse the impact of LUC, we distinguish between the grid cell temperature contribution and the grid cell temperature effect (see Sect. 2). The metric $\Delta T_{bgc}$ shows the effect on near-air surface temperature stemming from changes in land $CO_2$ fluxes due to historical LUC. In addition, we quantify how local land $CO_2$ fluxes due to LUC contributed to the global temperature change, quantified as $\Delta T_{bgc}^{local}$ (Fig. S8), computed using Equation (6). These maps do not show warming or cooling in the individual grid cells, but instead, if a grid cell contributed a warming or cooling effect to the global signal; this perspective becomes
relevant when considering deploying LUC intentionally to mitigate global warming, as is the case e.g. for reforestation. The multi-model mean (Fig.S8a) indicates an overall warming contribution, with only a few grid cells in the eastern U.S. and Europe showing a cooling contribution. This cooling contribution across the U.S. and Europe is primarily due to decades of reforestation and effective land management and underscores the potential of LUC as a CDR strategy. Historical records reveal that LUC, particularly reforestation, has the potential to provide the intended cooling benefits on global temperatures.
This historical precedent suggests that current and future LUC initiatives, such as A/R, could be effective in mitigating global warming, as evidenced by their cooling contributions over these regions. The variability across model estimates (Fig. S8b) not only suggests a dispersion in potential impacts of LUC-based mitigation strategies but also mitigates the risk of locking decision makers in a single outcome.



**Figure 5.** Time series of the global (**a**) and regional (**b**) - (**g**) response of near-surface air temperature due to biogeophysical effects of LUC ($\Delta T_{bgp}$) across 13 Earth System Models (ESMs). Results are computed as the difference between the *historical* and *hist-noLu* simulation from 1850-2014 for the global estimate in panel (**a**) and 1985–2014 for the regional estimates in panels (**b**) - (**g**). A 10-year running average is applied across both global and regional estimates. The thick black line and the grey shaded area in (**a**) represent the ensemble mean estimate and the standard deviation, respectively across all ESMs. The dash-dotted line represents the zero line. The acronyms are NATL = North Atlantic, NAM = North America, EURA = Eurasia, SEB = South East Brazil, WAFR = West Africa, SEA = Southeast Asia. Refer to Fig. 4a for the spatial extents used in computing panels (**b**) - (**g**).



We also analyse the local contribution of each grid cell to the BGP-induced global temperature change, quantified as $\Delta T_{\mathrm{bgp}}^{\mathrm{local}}$
(Fig. S9), computed using Equation (7). Our results show a warming contribution across the tropics including eastern Canada
and central Australia, whereas a cooling contribution dominates over the U.S. and Eurasia (Fig. S9a). Regions with a warming
contribution also correspond to high inter-model spread (Fig.S9b), whereas variability is lower over regions with a cooling
contribution except for the eastern U.S. Nevertheless, the ESMs again agree reasonably well in the direction of the grid cell
contribution to the global temperature change (Fig. S9d), with a pattern dominated by a cooling contribution, which polewards
switches to a warming contribution. Different from $\Delta T_{\mathrm{bgc}}^{\mathrm{local}}$, the $\Delta T_{\mathrm{bgp}}^{\mathrm{local}}$ cannot easily be interpreted as the contribution of the
LUC in a given grid cell to the global temperature signal. While the underlying carbon stock changes in $\Delta T_{\mathrm{bgc}}^{\mathrm{local}}$ are obviously
driven foremost by the LUC in that grid cell, the BGP temperature change in a grid cell results from the climate impacts of
local surface property changes as well as from energy and water vapour anomalies transported into the grid cell from LUC in
other locations. The pattern of $\Delta T_{\mathrm{bgp}}^{\mathrm{local}}$ is, therefore, a mixture of local and non-local effects of LUC (Winckler et al., 2019a),
and the two effects cannot be separated without additional simulations. Yet for regions of strong LUC (Figs. S13 - S16), the
local BGP effects appear to dominate. It is in this sense that our maps provide some guidance on the unintended effect of LUC
in a specific location on global climate via BGP pathways—which again may be indicative of LUC deployed intentionally to
dampen climate change.

We sum the BGC and BGP contribution to global temperature change to highlight the local contribution to the overall global
temperature change due to LUC (Fig. 6a). We emphasise that this does not correspond to any observable measure, but instead is
a metric for the relevance of a grid cell for the observed global temperature change in relation to the other grid cells. Although
our results focus on a multi-model mean, the BGC contribution of LUC dominates over the BGP contribution; this balance is
not spatially homogeneous. In the direction of signals, we find the warming contribution over the tropics as common across
the BGC and BGP effects (as in Fig. 2b of Windisch et al., 2021) but with opposing signals over the U.S. and Eurasia. In
magnitude, the warming pattern around Greenland can only be seen in the BGP contribution, which we attribute to mechanistic
non-local LUC-induced effects on ocean currents and sea ice. A few patches of grid cells towards the Arctic and grid cells
over the tropics, including parts of North America, contributed to warming, with lower warming from the former, while a few
grid cells over the U.S. and Europe contributed to a cooling of the climate. The spatial pattern of the combined effect (Fig. 6a)
resembles that of the BGC contribution (Fig.S8a) except for a more pronounced warming in the tropics. Overall, the cooling
contribution from the BGP effect is dampened by the warming contribution from the BGC effect.



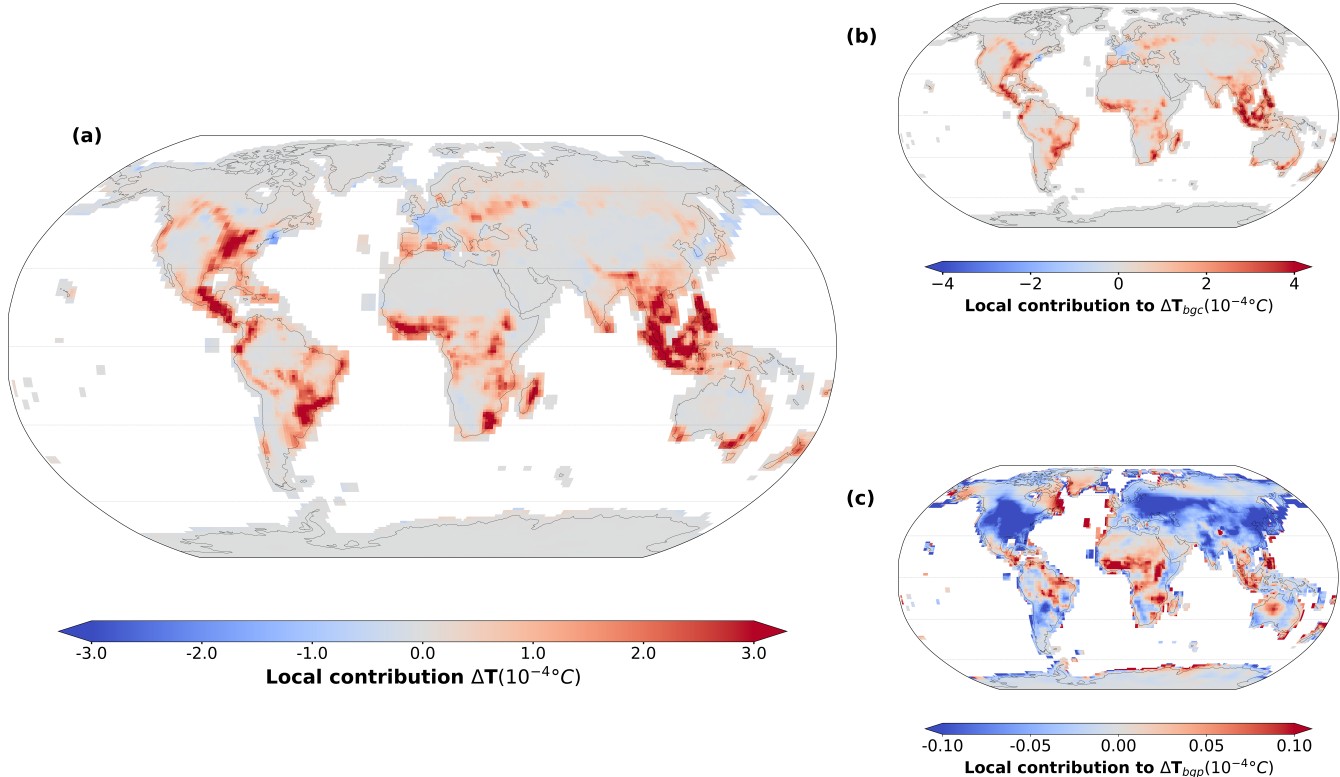

**Figure 6.** (**a**) Combined biogeochemical (BGC) and biogeophysical (BGP) contribution to global temperature change computed from the sum of the local BGC (**6b**) and BGP (**6c**) contribution computed using Equations (6) and (7) respectively, (**b**): local BGC contribution of each grid cell to global temperature change computed across 11 Earth system models (ESMs) as the product of the mean grid cell land-use emissions over 30 years (1985–2014) and the model-specific TCRE value, and (**c**) local BGP contribution of each grid cell to global temperature change computed across 13 ESMs, as the product of the mean grid cell temperature over 30 years (1985–2014) and the grid cell weighted area. Both BGC and BGP contributions are computed from the difference between the *historical* and *hist-noLu* simulation. Panels (**b**): and (**c**) are identical to Figs. S8a and S9a, respectively and have been reproduced here for the purpose of comparison. The ocean surface is masked out in panel (**c**) to isolate the local contribution resulting only from land surfaces. Refer to Fig. S9a for the full local BGP contribution including over ocean surfaces.



## 4   DISCUSSION

### 4.1   Disparity in estimates of near-surface air temperature across ESMs

Table 2 presented the estimated temperature responses from both BGP and BGC effects of LUC and compares them with results of other studies (Fig. 7). The temperature range due to BGC effects spans from -0.03 (BCC) to +0.40 °C (CESM2) across ESMs. The multi-model mean of 0.20 (±0.14) °C that we described earlier is similar to that of the IPCC estimate of 0.20 °C (Jia et al., 2019). Our estimate deviates only slightly from values retrieved from similar studies reporting global warming due to the BGC effects of LUC. In earlier studies based on single models, Brovkin et al. (2004) and Pongratz et al. (2010) estimated global warming of 0.18 °C, while Matthews et al. (2004) and Simmons and Matthews (2016) using different versions of the same model reported slightly higher global warming of 0.3 °C and 0.22 °C, respectively. Most recently, Devaraju et al. (2022), using an earlier version of CESM (CESM1) reported 0.24 °C in global warming due to BGC effects, which is about 40% lower than our estimate using CESM2 (0.40)°C. Our LUC-induced warming estimate is however likely underestimated, as we quantify the BGC effects of LUC based on the *1pctCO2* simulation (see Sect. 2). The *1pctCO2* simulation runs with pre-industrial land cover and does not consider the loss in forest area due to deforestation. As forests act as carbon sinks, a reduced forest area would increase the fraction and amount of $CO_2$ remaining in the atmosphere, thus causing larger warming.

The BGP effect range of -0.23 °C to +0.14 °C across models with a mean (standard deviation) of -0.03 (±0.10) °C that we described above is similar to an earlier global cooling estimate of -0.03 °C reported by Pongratz et al. (2010) and is also quite close to the estimate of -0.05 °C reported by Davin et al. (2007) in studies involving single ESMs. Our estimated mean and range are, however, substantially weaker than most estimates from previous studies (Fig. 7a), such as a BGP response range of -0.13 °C to -0.25 °C reported by Brovkin et al. (2006) in an intercomparison study involving six ESMs of intermediate complexity (EMIC), as well as estimates from other single-model studies yielding global effects of -0.06 °C to -0.22 °C (Matthews et al., 2004), -0.26 °C (Brovkin et al., 2004), and -0.1 °C (Lawrence et al., 2012) in global BGP effects. The large disparity across ESMs estimates of BGP effects, in both pattern and magnitude (Figs. 4 and S7), is not entirely unexpected as this has been reported in previous studies as well, including an intercomparison study involving 15 EMICs by Eby et al. (2013). Using seven LUCID atmosphere–land surface models (LSMs), De Noblet-Ducoudré et al. (2012) estimated a global cooling ranging from -0.005 °C to -0.056 °C. This spread was larger when models were forced with LUC than the combined effect of GHG and sea surface temperature due to large differences across ESMs regarding how the land cover type partitions the available energy. This leads us to speculate that the different implementation of LUC across models as seen across the different vegetative fractions (see Figs. S13 - S16) at least partly accounts for the widespread across the BGP estimates in this study. For example, tree cover fraction (Fig. S13), which is seen to vary considerably across ESMs, significantly influences surface temperature through mechanisms involving energy balance, albedo, and evapotranspiration. Depending on the location, forested areas, with their lower albedo, absorb more sunlight, leading to higher temperatures compared to lighter, non-forested areas that reflect more solar radiation. However, forests also have higher rates of evapotranspiration, which cools the air, and greater heat



capacity, moderating temperature fluctuations. As suggested by the multi-model mean of our BGP estimates (Figs. 4a), there is a possibility that historical LUC has caused feedbacks in sensitive components of the Earth system, namely Arctic and Antarctic

sea-ice, and may have influenced the Atlantic meridional overturning circulation (AMOC). Certain regional patterns, such as cooling over the U.S. Great Plains and Eurasia, are captured by most of the ESMs (Figs. 4 and S7), supporting earlier reported overall cooling over North America (-0.44±0.4 °C) and Eurasia (-0.3±0.3 °C) by De Noblet-Ducoudré et al. (2012), including other model-based (Boysen et al., 2020) and observation-based (Luo et al., 2022) deforestation studies. The cooling over the mid-latitudes has been reported as potentially driven by changes in both surface albedo and the surface moisture balance,

leading to increased latent heat flux and decreased sensible heat flux, especially in regions where crops were exchanged for short grass (Diffenbaugh, 2009; Mahmood et al., 2014; Chen and Dirmeyer, 2020; Bromley et al., 2020). The role of surface fluxes on climate and their dependence on soil moisture has been substantiated by other studies as even stronger over irrigated areas (e.g., Seneviratne et al., 2010; Thiery et al., 2017, 2020) as shown in the models simulating irrigation: CESM2 and the EC-Earth3 models (Fig. S7). In contrast to cooling over the U.S. Great Plains is the warming around Greenland's coast (Fig.

4a), is likely related to the coupled sea-ice-ocean feedbacks following increased export of Arctic sea ice into the subpolar North Atlantic described by Arellano-Nava et al. (2022). A combined decomposition of moisture flux convergence and surface energy balance analysis could be performed to investigate the source of these patterns, particularly over the higher latitudes as commonly simulated across the ESMs used in this study.

In assessing how carbon emissions in each grid cell due to historical LUC contributed to the estimated global BGC-induced

temperature change ($\Delta T_{\mathrm{bgc}}^{\mathrm{local}}$), the mean across ESMs indicates an overall warming pattern, with a higher magnitude of contribution over the tropics, particularly over southeast Asia (Fig. S8a). In a recent LUC assessment over southeast Asia involving TRENDY models, Kondo et al. (2022) attributed this change to also vary from peak in LUC emissions to over-dependence in forest products in the 1990s, which was countered by forest and environmental policy in the 2000s and beyond. Only a few grid cells in the eastern U.S. and Europe show a cooling contribution (Fig.S8a). The observed cooling contribution over Europe is

already well corroborated by Ganzenmüller et al. (2022) and also recently by Winkler et al. (2023) in what they attribute to changes in land use and land management primarily through land abandonment. Spatially, the mean warming pattern is largely coherent across models (Fig. S8d), but there exist some regions of larger model spread, notably over the continental U.S. and southeast Asia expressed by the inter-model variability (Fig. S8b) suggesting variability in potential impacts of LUC-based mitigation strategies but also mitigates the risk of locking decision makers in a single outcome.

Our findings indicate that the BGP effects ($\Delta T_{\mathrm{bgp}}^{\mathrm{local}}$) have resulted in a warming contribution across the tropics, including regions like eastern Canada and central Australia, while cooling contributions are more prevalent over the U.S. and Eurasia (Figs. 6a and S9a). The warming contribution over the tropics is mostly attributable to the latitudinal impact of deforestation and is already well corroborated across previous research studies, which gives confidence to our results. For example, the results of idealised deforestation studies by Li et al. (2022b) and Boysen et al. (2020) revealed that the BGP effects of deforestation,

such as reduced precipitation and increased temperature, could amplify carbon losses with the resulting regional warming having stronger impacts on tropical ecosystems than warming from global radiative forcing. Zhu et al. (2023), using idealised



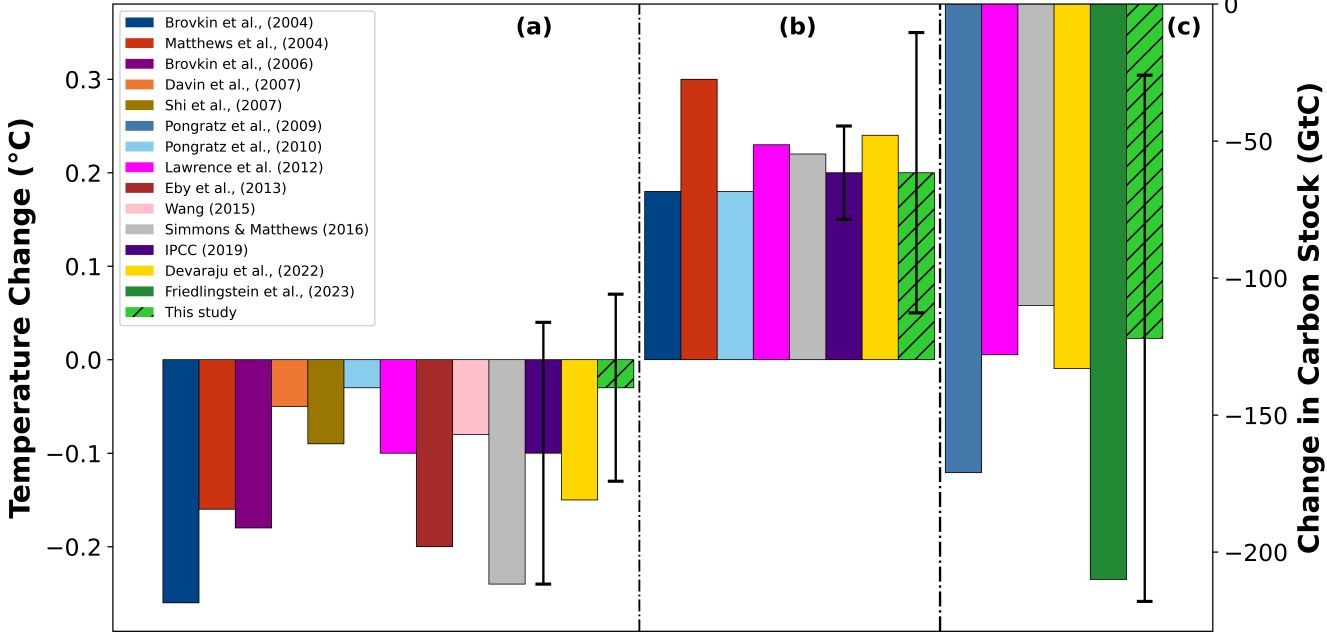

**Figure 7.** Biogeophysical (**a**), biogeochemical (**b**) effects, and changes in carbon stocks (**c**) quantified in this study (hatched green bars) compared with other studies. Where vertical lines exist, they represent the standard deviation of estimates. See Supplementary Table S2 for the studies and their estimation periods.

deforestation scenarios, also showed that deforestation in the Amazon results in significant local warming and drying with a substantial reduction in rainfall, exacerbating temperature increases. Similar patterns were also observed in the Congo, where deforestation led to local temperature rises due to decreased precipitation and increased dryness. This was also supported by

Zeng et al. (2021) in a study across tropical mountain regions, which found that tropical deforestation led to an increase in surface air temperature due to decreased evapotranspiration and changes in albedo, with a notable elevation dependency: higher elevations experienced cooler temperatures, while lower elevations were warmer. Similarly, Windisch et al. (2021) assessed the impact of climate mitigation policies and demonstrated that the conservation and reforestation of tropical forests provide the highest climate benefit by significantly reducing local temperatures. They also found that these measures can lead to local

warming at higher latitudes during winter, analogous to the latitudinal and also elevation dependence observed by Zeng et al. (2021). These demonstrate the unintended effect of LUC in a specific location on global climate via BGP pathways—which again may be indicative of LUC deployed intentionally to dampen climate change. Given that future climate might differ from the past, the BGP effects of the same LUC may change. As the climate warms, the influence of LUC on surface temperature could become more significant. Winckler et al. (2017) identified key factors: afforestation scenarios like RCP4.5 tend to cool

surfaces, while deforestation causes warming, driven by changes in albedo, energy balance, and heat fluxes. Regional impacts vary; for example, forest die-back in the Amazon (particularly under RCP8.5) resulted in a cooling effect, while the northward




shift of the boreal tree line induced warming (Winckler et al., 2017). Deforestation starting from lower forest fractions leads to more significant warming, especially in a warmer climate due to reduced snow cover and changes in heat fluxes (Pongratz et al., 2011). Human activities, including GHG emissions and LUC, have been shown to warm the mid-latitudes more than

the tropics, with higher $CO_2$ levels increasing precipitation and intensifying the hydrological cycle. Pitman et al. (2011) also showed that changes in snow and rainfall under increased GHGs dominate how LUC affects regional temperatures. Such changes would impact the snow-albedo feedback and water supply, limiting evaporation and controlling LUC's net climate impact. Buechel et al. (2024) recently found that while afforestation has some detectable effects on regional hydrology, these effects are small compared to the more substantial impacts of variables such as precipitation, temperature, and $CO_2$ levels. In

regions such as Eurasia and the eastern U.S., increased $CO_2$ results in higher precipitation and moisture availability, enhancing the cooling effects of LUC. The poleward cooling contribution towards the higher latitudes also seen in our study has also been reported by De Noblet-Ducoudré et al. (2012).

In our attempt to highlight the combined contribution to global temperature change due to historical LUC, we show the aggregate of the BGC and BGC effects as an overall warming contribution (Fig. 6a). The BGC effects, felt globally as the $CO_2$

released by LUC mixes in the atmosphere (Grant et al., 2023; Ito and Hajima, 2020; Pongratz et al., 2021), contrast with the often globally negligible BGP effects, which, as earlier mentioned, exert a stronger influence on climate at the local scale. The interaction between BGP and BGC effects is revealed to result in complex climate impacts, with BGP effects either mitigating or enhancing BGC-induced warming over different regions (Pongratz et al., 2021; Windisch et al., 2021; Jia et al., 2019). However, depending on the geographical location, BGC effects are shown to also amplify the warming caused by local BGP

effects (Windisch et al., 2021; Boysen et al., 2014). While the strong warming pattern over the tropics can be traced to both the BGC and the BGP effects, the poleward warming contribution is due to the BGP effect alone, which includes both the local and non-local effects of LUC. There is a range of evidence that non-local BGP effects, as a teleconnective consequence of LUC occurring elsewhere (Pongratz et al., 2010), regionally dominate over local BGP effects. This has been demonstrated across deforestation experiments (e.g., Winckler et al., 2019b; Davin and De Noblet-Ducoudré, 2010) and most recently in an

idealised LUC experiment (De Hertog et al., 2023). While the global mean BGC effect is shown to dominate over the BGP effect, we also observe that the BGP effects are more relevant on a regional scale than suggested by the global mean. Our result, therefore, reaches a similar conclusion as Pongratz et al. (2011), which demonstrated the global dominance of $CO_2$ over albedo forcing, contributing to warming and cooling, respectively, albeit with regional specificities. Understanding the regions where these effects differ in magnitude and direction could help in the attribution of historical climate change. For instance,

research has shown that historical warming can be attributed to human activities beyond changes in GHGs (Bruhwiler et al., 2021; Hegerl et al., 2007), including aerosols (Seinfeld et al., 2016), land use (Hegerl et al., 2007), and changes in the Earth's energy absorption and reflection (Bruhwiler et al., 2021). Anderson et al. (2016) highlighted additional climate system feedbacks, such as the melting of snow and ice, which alters albedo, and reduced land carbon uptake in a warmer world (Solomon et al., 2010). Therefore, regardless of the global dominance of the contribution from the BGC effects, the role of the contribu-

tion from the BGP effects on both regional (local effects) and global (local + non-local effects) climate cannot be overlooked.



For local mitigation and adaptation projects, a separation between local and non-local effects will be crucial. While the BGP effects of land-based CDR on global temperature is still subject to ongoing research, such effects are reported to depend on the scale and type of CDR deployment and resulting modification of the Earth's surface energy balance. In addition to the potential of land-based CDR techniques such as A/R, bioenergy crop cultivation and soil carbon sequestration practices to

alter surface characteristics like albedo, energy partitioning, evapotranspiration, and surface roughness (Bonan, 2008; Jackson et al., 2008; Betts, 2000; Buechel et al., 2024), these modifications could lead to potential global and regional temperature changes (Cheng et al., 2024; Windisch et al., 2021; Cerasoli et al., 2021), and in some cases even beyond where the LUC is implemented (De Hertog et al., 2023; Winckler et al., 2019a). Such changes in BGP processes can impact local and potentially global temperatures, with effects shown to vary with latitude and regional characteristics; such as instances where, reforestation

leads to decreased albedo and increased evapotranspiration, affecting cloud cover and regional temperatures with latitudinal dependence (Bright et al., 2017; Arora and Montenegro, 2011). Similarly, agricultural techniques that enhance soil carbon sequestration or the use of bioenergy crops have been reported with the potential to alter local climate through changes in albedo and surface roughness (Hirsch et al., 2018; Davin et al., 2014). Zickfeld et al. (2023) suggested a continental or global-scale implementation of land-based CDR techniques would be necessary for significant global temperature modulation, but the

results of the combined effects suggest that such impact might be more visible at local or regional scales of implementation.

## 4.2   Regional variability in BGP vs BGC effects on near-surface air temperature

To better understand the robustness of our estimates, we closely examine the BGP and BGC effects on near-surface air temperature across selected regions (see Sect. 3.3). The multi-model mean shows a wide spread in $\Delta T_{bgp}$ estimates, especially in regions with strong effects like NAT and NAM (Figs. 8a and c). This variability is due to diverse responses to LUC and

different interpretations of surface flux changes by atmospheric models. The high variability in BGP effects is due to immediate, heterogeneous local changes in land cover, affecting surface heat fluxes and albedo (Duveiller et al., 2018). Boysen et al. (2014) also noted significant BGP effects in regions with intense LUC, with varied responses depending on the model and region. In tropical regions, changes in latent heat fluxes are more impactful than albedo changes, while in mid- to high-latitude regions, seasonal changes in albedo due to snow cover are influential. Our estimate of the LUC-induced BGP effects is however

likely underestimated, as we employed monthly temperatures and thus, did not account for the difference between daytime and nighttime temperature. The difference in impact between daytime versus nighttime temperatures has been shown to differ in directional impact (Lejeune et al., 2017; Qiao et al., 2013; He et al., 2024) and can often average out to near-zero impacts. Sub-diurnal temperatures were also recently demonstrated to have a "reversed asymmetric warming" due to other factors beyond LUC (Zhong et al., 2023). Although beyond the scope of our analysis, we highlight that accounting for the daytime vs

nighttime temperature difference could lead to diverging estimates from those presented in this study. The BGC effects, on the other hand, lead to more uniform temperature changes (Figs. 8b and d) as they are influenced by well-mixed GHGs, nutrient cycling from different vegetation types, soil composition, and human activities like agriculture and deforestation (Pongratz et al., 2010). These processes contribute to a more consistent and homogeneous impact on regional and global climates. Our



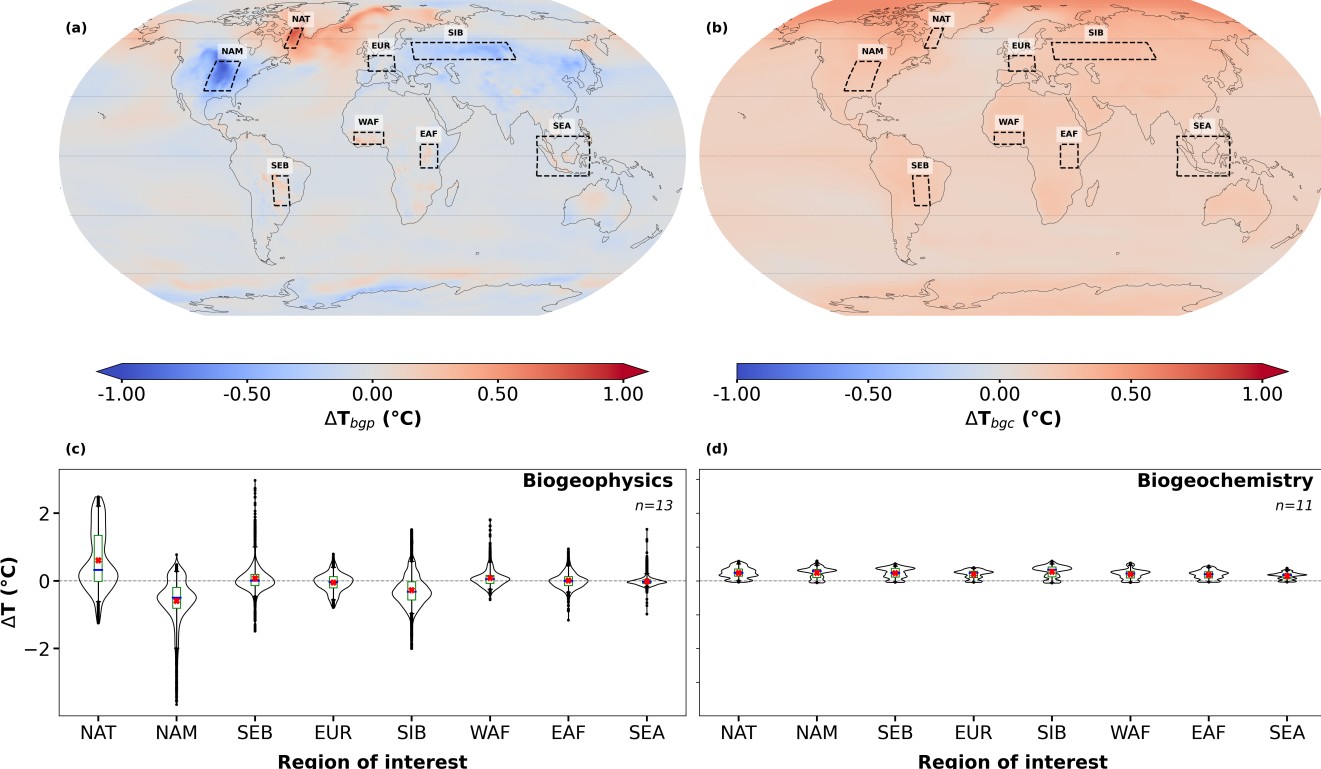

**Figure 8.** Inter-model variability across ensemble average in near-surface air temperature $\Delta T$ due to (**a**) biogeophysical effects ($\Delta T_{bgp}$) and (**b**) biogeochemical effects ($\Delta T_{bgc}$) of land-use change, computed as the mean of 13 and 11 Earth system models (ESMs), respectively. (**c**) and (**d**) indicate the spread in model estimates of $\Delta T$ for the regions highlighted in (**a**) and (**b**). The violin plot shows the distribution shape of estimates indicating probability. The boxplot bounds the 1st and 3rd quartiles (25th and 75th percentiles, respectively) with the red X and solid blue bar (-) representing the mean and median estimate from all grid cells within the region of interest, respectively. The vertical whiskers extend from the 1st to the 3rd quartiles (the upper and lower boundaries), while the outliers are represented by *. For ease of reference, Figs. 4a and 3a have been reproduced here in panels (**a**) and (**b**), respectively.

findings further demonstrate the complexity and variability of BGP effects on regional climates, emphasising the need for care-
ful consideration of local LUC effects and their direct impact. In contrast, the more consistent BGC effects reflect the influence
of well-mixed GHGs, leading to more uniform temperature changes. Integrating diverse regional responses into global models
might be a step in the right direction towards improving the accuracy of climate projections and region-specific mitigation and
adaptation strategies.





## 4.3 Variability across models' estimates of land-use change carbon emissions

Historically, LUC has had pronounced effects on carbon distribution over land surfaces across the ESMs. The multi-model mean change in carbon due to historical LUC of -122±96 GtC reported earlier is close to LUC estimates of -110 GtC, -128 GtC, and -133 GtC reported by Devaraju et al. (2022), Lawrence et al. (2012), and Simmons and Matthews (2016), respectively, but substantially lower than other estimates (Fig. 7c and Table S2). Besides the -210 GtC of cumulative emissions from LUC estimated by GCB2023, earlier estimates fall within the upper uncertainty bounds of our reported multi-model mean carbon

loss. Our results indicate that the BGC effects of LUC are not only consistent across ESMs but with spatial homogeneity over certain regions. For example, there is a consistent depiction of changes in carbon storage in the Northern Hemisphere, especially over the boreal forests in Canada and Russia, which could be indicative of shifting forest boundaries, logging, or other disturbances in addition to factors such as LUC or pest infestations reported by Foster et al. (2022). Notably, over Europe, we observe an increase in carbon stocks, designating Europe as a carbon sink. In addition to an array of studies (e.g.,

Kilpeläinen and Peltola, 2022; Pilli et al., 2022; Lasanta et al., 2017), the observed increase across European carbon pools was reported by Ganzenmüller et al. (2022) who largely attributed the increasing carbon stocks to agricultural land abandonment, reforestation, and forest regrowth among other drivers (e.g., Fayet et al., 2022; Perpiña Castillo et al., 2021). The degree of change near the Arctic region also shows clear variations. Over the Northern Hemisphere, several models indicate carbon gains in parts of North America, Europe, and northern Asia. This is due to afforestation, reforestation, reduced cropping, and/or

natural forest regrowth in these regions (Figs. S13 and S14). In examining the effects of land-use change on the U.S. Great Plains, the divergences in $\Delta$cSoil values observed in BCC and CESM2 models provide additional insights. This discrepancy underscores a finding from Derner et al. (2019), who, in their study on soil carbon sequestration across the Great Plains, concluded that grazing does not significantly contribute to variations in soil carbon levels; claims that were recently contested by Ren et al. (2024). We find no relationship between $\Delta$cSoil and models that implement grazing (Table 1). In parts of Western

Australia, there's a general agreement (Figs. 2 and S2) among models indicating carbon loss, likely reflecting land use changes such as deforestation and agricultural activities. In contrast, the evident decline in total land carbon over regions including the Amazon rainforest, western and central Africa, and Southeast Asia is consistent with the widely reported tropical deforestation (Smith et al., 2023; Zeng et al., 2021; Lejeune et al., 2018).

Our results of the spatial distribution of both increase and decline in total land carbon is supported by previous model and

observation studies, attributing the decline in the tropics to deforestation and land degradation (Zhu et al., 2023; Matricardi et al., 2020; Baccini et al., 2017; Lorenz et al., 2016), regrowth following historical wood harvest and deforestation over the U.S., Russia, and Europe (Pongratz et al., 2008), and expanding forest area/regrowth due to abandonment/forest transitions in the 19th and 20th century over Europe (Ganzenmüller et al., 2022; McGrath et al., 2015). Our observed carbon gain across Europe is also well corroborated by Ganzenmüller et al. (2022) and Winkler et al. (2023) in what they attribute to changes

in land use and land management primarily through A/R and land abandonment. We find a general decrease in $\Delta$cLand over Southeast Asia (Figs. 2 and S2), though with large model-to-model uncertainty (Fig. 2b) supporting the findings by Kondo et al.



(2022). In their analysis involving TRENDY simulations, some of which also serve (though partly in older model versions) as LSMs of some ESMs used in this study, variability across model estimates of LUC fluxes over Southeast Asia was partly due to incomplete processes in the LUC forcing data, including temporal changes in peatlands conversion and the overlooked

carbon cycle of oil-palm ecosystems. Kondo et al. (2022) also attributed this variability to peak LUC emissions resulting from heavy reliance on forest products in the 1990s, which was later mitigated by forest and environmental policies in the 2000s and beyond. The revealed variability across models underscores the inherent uncertainties associated with modelling complex Earth system processes in addition to model input parameterisation.

Furthermore, the implementation of gross versus net transitions of LUC is seen to influence the resulting carbon emissions.

Specifically, our estimates of LUC carbon fluxes reveal that ESMs implementing gross transition (i.e., bidirectional land-use changes within a grid cell, e.g. forest to crop and crop to forest) result in higher estimates compared to models implementing net transitions (i.e., overall changes in land use categories within a grid cell over a given time period, without accounting for the processes that occur within the grid cell). Of the 13 ESMs in the present study, all models (except for BCC) considering gross transitions, sit within the uncertainty range of the reference value of cumulative emissions due to LUC as estimated

by GCB2023. The gross versus net transitions issue usually relates to the inclusion of shifting cultivation areas in ESMs (Ganzenmüller et al., 2022; Lawrence et al., 2016) but has also been contested to occur everywhere on the globe (Fuchs et al., 2018). Including shifting cultivation is reported to lead to higher estimates of LUC carbon fluxes (Hartung et al., 2021; Stocker et al., 2014; Wilkenskjeld et al., 2014), increasing estimates by 20%–30%. Bayer et al. (2017) also noted that accounting for gross transitions significantly amplifies the impact of LUC on carbon stocks and fluxes releasing up to 15% more carbon

compared to when net transitions were considered. Furthermore, Bastos et al. (2022) highlights substantial differences due to the implementation of gross transitions in estimates based on LUH2 compared to estimates based on other LUC datasets. This suggests that the land-use transition structure within an LSM is likely to govern the model's ability to produce accurate land $CO_2$ fluxes relative to other system parameterisations. We find that the BCC and CESM2 models are the only two exceptions to the gross versus net transition issue discussed above. While the former, implementing gross transition, simulated carbon

gain due to LUC, the latter, implementing net transitions, shows high carbon loss, at a magnitude, comparable to models with gross transitions. Therefore, while the gross versus net transition issue can be generally used to explain the model response to LUC, further explanatory processes, singly or in combination, would be required to explain the model behaviour of BCC and CESM2.

Building on the above, CESM2 is also one of the three models (the other being the EC-Earth3 models) to implement irrigation,

which from our estimates (Table 2), consistently show high land $CO_2$ flux estimates. Irrigation has also previously been reported to induce LUC carbon flux estimates (Qin et al., 2024; Roy et al., 2022; Taheripour et al., 2013) beyond its BGP impact on the land surface (De Hertog et al., 2023; Al-Yaari et al., 2022; de Vrese and Hagemann, 2018). According to de Vrese and Hagemann (2018), this is largely because irrigation represents a type of heterogeneity that introduces very sharp contrasts in land surface characteristics, in that it increases water availability only in a certain fraction of the grid cell.



For ESMs simulating wood harvest (ACCESS, CESM2, CMCC, IPSL, GFDL, MIROC, and MPI), biomass due to harvest is transferred to the product pool (on- or off-site) and eventually decays into the atmosphere. The harvest residues are transferred into soil/litter carbon pools in all ESMs but CMCC (harvest goes into the product pool) and IPSL (no residues considered). Additionally for IPSL, wood harvest is considered once per year, while only aboveground biomass is harvested and put into product pools (Lurton et al., 2020). In MIROC, the occurrence of carbon harvesting depends on the transition pattern (e.g., the conversion from natural vegetation to cropland causes carbon harvesting; cropland or pasture abandonment does not imply any carbon removal). In contrast, for ACCESS, harvest only occurs when forest area decreases. The impact of neglecting wood harvest on estimates of the LUC flux is substantial. Wood harvest has been found to contribute between 19% (Stocker et al., 2014) and 28% (Hartung et al., 2021) of the cumulative LUC flux, highlighting its critical role in accounting for anthropogenic land-use emissions. According to Hartung et al. (2021), this contribution was larger than the impact of LUC flux uncertainty, which affects the cumulative LUC flux by up to 22%. This underscores the necessity of including wood harvest in ESMs to achieve more accurate carbon flux estimations.

Another major difference in LUC carbon flux estimates can be linked to the initial conditions (pre-industrial control) of the carbon cycle even across models with similar LUC implementation. We observe little-to-no relationship between the initial conditions and the global changes in the carbon pools (not shown). However, we find spatial correlation at the regional level. The variations in initial conditions of land cover and land use of ESM simulations is known to impact the estimates of global carbon cycle feedback parameters even under idealised scenarios. Using data from six LSMs that also serve as the LSMs for ESMs in this study, Boysen et al. (2021) found that initial soil organic carbon (SOC) stocks differ among LSMs due to different approaches in representing SOC. Using data from 10 ESMs participating in CMIP5 (six of the ESMs using earlier versions of the LSMs as those in this study), Exbrayat et al. (2014) also reported large differences in initial SOC stocks among models due to variations in decomposition processes during model setup. Tian et al. (2015) showed that in the first simulation year, global SOC stocks varied widely, while changes over the following years accounted for only a small percentage of the initial SOC stock. Using data from 10 terrestrial biosphere models, they reported a strong correlation between initial (year, 1901) and contemporary (year, 2010) SOC estimates, driven by significant differences in initial SOC stocks. Recently, Melnikova et al. (2022) noted that, even for the same land-cover types, variations in pre-industrial land covers among ESMs result in spatial differences in ecosystem carbon stocks (e.g., models with larger forest cover have larger land carbon pools). It is, therefore, conceivable that models with the same implementation of LUC can lead to order differences on the global scale simply because of the baseline conditions on which the LUC is imposed. Based on our estimates of net LUC carbon flux, we find that a wide margin already exists across model estimates; a range that also existed at the beginning of the CMIP5 historical simulations (Exbrayat et al., 2014). We also find a spatial correlation between the initial condition (year, 1850) and estimates at the end of the simulation (year, 2014). Specifically, across the spatial patterns, models like GFDL and CESM2 already show a substantial reduction in total land carbon at the beginning of the simulation (not shown). Since the ESMs used almost similar externally specified land-use forcing, LUH2, we resolve the large range in initial conditions to the internal model differences.



Despite the shared framework of the LUMIP protocol and LUH2 dataset, each ESM's unique architecture leads to considerable variability in outputs. This variation stems from different approaches to LUC implementation and carbon pool accounting,
ranging from nuanced LUH-based transitions to aggregated vegetative fractions. Such heterogeneity highlights the complexities of simulating land management and carbon dynamics (Boysen et al., 2021) and underscores the challenge of drawing general conclusions about land carbon emissions and temperature responses from an ensemble of ESMs. Consequently, this breadth of outcomes emphasises a critical point: despite underlying commonalities in data sources and objectives, the disparities in model implementations and results appear to dominate, suggesting that the differences in ESMs' treatment of land management
and carbon cycle processes may indeed outweigh their similarities. Furthermore, progressive but non-uniform inclusion of land management practices (see Blyth et al., 2021) is speculated to increase the model spread and thus divergence across model estimates (Pongratz et al., 2018). Such variability not only emphasises the need for cautious interpretation of model outputs but also signals a pressing need for increased coherence in model development. It also highlights the importance of advancing model intercomparison and harmonisation efforts needed to plug the gap between estimates from ESMs. Such
efforts will be fundamental to enhancing the reliability and comparability of climate projections crucial for informing global environmental policy and climate action strategies, particularly in the face of increasing land-use changes and their profound climatic implications.

## 5 CONCLUSIONS

Model intercomparison studies provide a multifaceted advantage, facilitating the computation of the mean response of mod-
els, quantification of associated uncertainties by examining the dispersion across different models, and illustrating the factors contributing to uncertainty. The analysis of land carbon dynamics within LUMIP permits exploring the progression of LUC representation in ESMs and related signal strengths since LUCID and CMIP5. In this study, we primarily focused on separating the climate response caused by biogeophysical (BGP) effects of historical land-use change (LUC) from those caused by biogeochemical (BGC) effects. We go beyond previous studies to analyse the most recent CMIP6 data, using state-of-the-art
datasets contributed by the LUMIP project in an attempt to improve existing knowledge on the relative contribution of BGP and BGC effects of LUC on climate.

We quantified the BGC effect of LUC on near-surface air temperature of 0.20±0.15°C to be globally dominant over the BGP effect of -0.03±0.10°C, which is in agreement with earlier studies. However, on the regional scale, the BGP effect is shown to be more important than suggested by its global mean. For example, while temperate and boreal regions contribute to BGP-
induced cooling, the tropical regions have contributed to BGP-induced warming during the historical period. The prevalence of BGP effects in climate analysis remains invaluable for attributing historical climate. As already evidenced by research, land-based CDR techniques, like reforestation and agricultural practices, can substantially alter surface characteristics like albedo, evapotranspiration, and roughness, thereby impacting regional and potentially global climate (Zhu et al., 2023; Windisch et al.,



2021; Boysen et al., 2020; De Noblet-Ducoudré et al., 2012). Although contingent upon the type and scale of CDR deployment
(Zickfeld et al., 2023), the role of BGP effects remains vital for local mitigation and adaptation policies. While the BGC-
induced effects exhibit a globally dominant warming effect, highest over the Arctic, we find this as the only commonality with
the BGP effects, which transition to a cooling strip on land surfaces over the mid-to-upper latitudes, from the U.S. Great Plains
to the Northeast Plain of Asia. Nuanced warming patches are also evident over the tropics and subtropics, but these signals are
largely mixed, indistinguishable, and difficult to attribute.

Additionally, we identified the historical effects of LUC on total land carbon ($\Delta$cLand) as consistent in the direction of change
across most models, with a mean and spread loss of -122$\pm$96 GtC. Notably, this differs from the -210$\pm$65 GtC estimate
reported in GCB2023. Our study reveals that historical LUC emissions estimated by coupled models are significantly lower
than those reported by other models, such as those used by the Global Carbon Budget. The bookkeeping models used in GCB
differ in their assumptions, methodologies, and data inputs from the ESMs used in our study. Coupled models, which account
for interactions between climate and land systems, may offer unique insights into land-use impacts on carbon emissions. The
discrepant estimates highlight the importance of using diverse modelling approaches to capture the full range of potential
impacts of land-use changes. Further investigation into these factors is necessary to reconcile these estimates, improve our
understanding of LUC impacts on land carbon dynamics, enhance the accuracy of carbon budget assessments, and inform
more effective climate policy strategies.

The contributing fluxes and impacts on specific carbon pools also differ strongly across models and regions, particularly due to
the interplay of vegetation cover and carbon pools. While it is possible to attribute trends in $\Delta$cLand to be largely dominated
by the contribution from vegetation carbon pools, this generalisation does not hold true for all ESMs. Despite using the latest
generation of ESMs, our results identify a wide variation in land $CO_2$ fluxes, which is largely traceable to the underlying
model architecture, particularly in how models implement LUC and land management using the LUH2 dataset. We also see
clear differences in the treatment of land management in ESMs, but also some common features. Primarily, what constitutes
total land carbon still differs widely across models based on designed model architecture to implement climate processes even
when using a similar modelling protocol. We find that the implementation of gross versus net land-use transitions at the subgrid
level also influences estimates of LUC carbon fluxes. Specifically, models implementing gross transitions resulted in higher
estimates compared to those implementing net transitions, in addition to other factors such as wood harvest and the initial
condition (pre-industrial treatment of land use) of LUC. The resulting range in estimates shows that improvement efforts are
needed to narrow the gulf between models to support more robust climate estimates, thus, joining earlier recommendations for
a better representation of biogeophysical (e.g., Duveiller et al., 2018; Forzieri et al., 2018) and biogeochemical (e.g., Schädel
et al., 2024; Bastos et al., 2022) processes in ESMs.

A novel aspect of our study lies in our attempt to estimate the grid-scale contribution to global temperature change. We show
the regional disparity in the BGC contribution to LUC-induced global temperature change as highest over the tropics and
subtropics where LUC was mostly registered. Patches of grid cells over the eastern U.S. and western Europe show a warming



contribution to global temperature. While a BGP-induced cooling contribution is more prevalent over the U.S. and Eurasia, we find warming contributions over regions such as eastern Canada, central Australia, and the tropics. We identified the warming contribution over the tropics resulting from the BGP effects as the only commonality between the BGP and BGC effects.

Treated in combination (BGP and BGC effects), we identify a much higher warming contribution from the North Atlantic attributable to BGP effects, including the local and non-local effects of LUC owing to the teleconnective consequences of LUC occurring elsewhere. Therefore, recognising and accounting for both local and non-local effects of LUC on climate is essential for developing holistic climate policies that address the full range of impacts associated with LUC. Such insights, crucial to identifying where land-based projects can potentially alter surface temperature, can enhance mitigation benefits at both the

local and global scales. Additionally, the heterogeneous nature of LUC also leads to impacts at scales often too small to be captured by the spatial resolution of most ESMs. Standard ESM output provides only grid cell average impacts. However, for the subgrid areas where the land cover change occurs, the effect might be significantly larger than suggested by the grid cell mean; a phenomenon also known as "hyper-local" impact. While these specifics are essential, their considerations were beyond the scope of our analysis. We therefore highlight that accounting for them, when possible, could lead to diverging estimates

from those presented in this study.

Finally, our estimates of both BGP and BGC effects still differ from estimates in other studies albeit agreeing in the direction of change (BGP: cooling, BGC: warming). Unlike the BGP effects, our estimated BGC-induced warming is close to previously reported values including estimates from previous generations of models. We anchor our findings on the premise that BGC fluxes typically rely on an ensemble of models for a robust best estimate (e.g., Friedlingstein et al., 2022a, b) despite the

difference in implementation of land-use and land management practices across models. Therefore, in concluding this model intercomparison study, we note that an ensemble of the latest generation of ESMs produces a mean and spread in estimates for both BGP and BGC effects that are large and similar to that seen across previous LUCID and CMIP5 estimates. We have highlighted a few aspects in the model architecture contributing to the observed spread but also note that ESMs currently stand at different progressive stages with some accounting for more land-use and land management processes than others. As

models strive to become more complete by implementing more processes, such improvements could likewise lead to more divergence across model estimates even with consistent anthropogenic forcings. Convergence can most likely be anticipated when modelling groups are able to achieve reasonable implementation of the major land-use processes; efforts that will be pivotal for and enhance confidence in future climate projections.



*Data availability.*

The data and scripts used are available upon request from the corresponding author.

*Author contributions.*

JP initialised the project and provided the research idea. AAA, CS, and JP designed the study. AAA performed the analysis and wrote the manuscript with inputs from CS and JP. All authors contributed to evaluating the results and proofread the manuscript.

*Competing interests.*

The authors declare no conflicting interest but also wish to highlight that at least one of the (co-)authors is a member of the editorial board of Earth System Dynamics.

*Acknowledgements.* This work used resources of the Deutsches Klimarechenzentrum (DKRZ) granted by its Scientific Steering Committee (WLA) under Project ID mj0060. We thank the climate modelling groups for producing and making available their model output, the Earth
System Grid Federation (ESGF) for archiving the data and providing access, and the multiple funding agencies that support CMIP6 and ESGF.



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
