# Peer review of "Biogeochemical versus biogeophysical temperature effects of historical land-use change in CMIP6"

_EGUsphere, 2024_

## Author Comment (AC1)

**Response to Reviewer #1**

**General comments**

In their aptly titled 2024 article "Biogeochemical versus biogeophysical temperature effects of historical land-use change in CMIP6," authors Amali et al. quantify the biogeophysical (BGP) and biogeochemical (BGC) effect of historical land-use change (LUC) as rendered in 13 earth system models of the sixth Coupled Model Intercomparison Project's (CMIP6) Land Use Model Intercomparison (LUMIP) activity. Specifically, the authors seek to analyze the effects of historical LUC for carbon emissions and near-surface air temperature. Although the relative contributions of BGC and BGP effects of historical LUC have been studied using CMIP5 and Land-Use and Climate, Identification of Robust Impacts (LUCID) data, CMIP6's LUMIP activity, prescribes a set of experiments to be carried out in common by modeling teams, using the latest generation of earth system models. The study is timely as the BGP impacts of LUC have often been overlooked. Where it has not been overlooked results have at times been difficult to interpret due to the variety of LUC schemes applied within CMIP5. This study avoids this particular challenge by using data from the latest generation of models and experiments where simulation protocols dictate greater consistency across models.

Two concentration-driven CMIP6 simulations are used by Amali et al. to analyze the effects of historical LUC. The *historical* simulation with LUC from 1850 to 2015 and *hist-noLu* where LUC is held constant from 1850. The difference between the two simulations is taken to determine the change in carbon storage and near-surface temperature. The authors use the TCRE to find the BGC temperature effect of LUC. To obtain gridcell depictions of this temperature effect, the authors use the regional-to-global ratio of temperature (or simple pattern scaling). These methods allow the authors to isolate the impact of historical LUC on the variables of interest and identify the contributions of BGC and BGP for each.

The study's findings both align with and expand upon previous work. For example, the finding that near-surface temperature increase from BGC is greater than BGP for historical LUC aligns with the findings within the existing literature. However, the regional analysis in Amali et al. adds nuance to this story in that the regional effect of BGP on near-surface temperature can be significant depending on location. Also significant is the study's contribution to our understanding of the BCG effect on near-surface temperature change at the gridcell level. Furthermore, the findings of this study demonstrates similar model spread and estimates to previous similar studies using LUCID or CMIP5 data, and identifies some reasons related to model architecture that contribute to this result.

This study is ambitious in scope, well-referenced, and contributes significantly to our understanding of the relative temperature contributions of the BGC and BGP effects of LUC, using a novel RGRT approach to do so. Its conclusions are supported by the results, however, it's possible that the conclusion that both the local and non-local effects of LUC ought to be considered in climate policy development should be qualified, noting that this is because combined local and non-local BGP effects of LUC found in this study are not insignificant. The article is recommended for publication pending consideration of the questions and comments that follow.

**Response:** *Thank you very much for taking the time to review the manuscript thoroughly and for providing valuable feedback. We are glad for your positive and constructive evaluation of the manuscript, recognising its relevance, novelty, and well-referenced nature. In the following sections, we try to provide a point-wise response to the specific and technical comments raised.*

**Specific comments**

1. The abstract provides a complete and concise summary.
2. The manuscript is also well-structured in that the sections and subsections allow the authors to present their methods, results, and discussion in a manner that is both logical and appealing from a reading flow perspective. One subsection that might benefit from being split in two is 2.3.2, where "Global temperature response" and "Local contributions to global temperature change" could each be their own sub-sections.

   **Response:** *Thank you for pointing this out. We have now restructured the manuscript following your suggestion. The revised structure of the subsections now reads,*

   > *2.3.2: Global temperature response*
   > *2.3.3: Local contributions to global temperature change*
   > *2.3.4: Descriptive Statistics*

3. The figures do a good job of presenting the results and key points for discussion in a readable fashion. Related to the methods and the results presented in Figure 3 where *ΔTbgc* is presented, why is there no test of statistical significance as is the case for *ΔTbgp*?

   **Response:** *Thank you for pointing this out. We have now conducted a 1-sample t-test on the 1pctCO2 simulation. For the multi-model mean $\Delta T_{bgc}$ (Fig. 3), stippling indicates regions where 2/3 of the models are not statistically significant at the 95% confidence level. We have added this information to Fig. 3a. The result of the individual models is shown in the figure below, which we have now updated for Figure S6. We have also updated the methods section to reflect this.*

[Figure]

***Figure S6:*** *Temperature response to land-use-induced CO2 fluxes. Results computed from Equations 2 and 3 using global mean land-use CO2 emissions (1985 – 2014), global mean temperature from 1pctCO2 simulation, and TCRE values derived in Arora et al. (2020) and Lovato et al. (2022). Stippling indicates regions that are not statistically significant at the 95% confidence level.*

4. The methods are clearly described, including useful model information presented in tabular form and details of the statistical analysis. Related to the latter, is it possible to include which type of interpolation method was used to bring all of the simulation data into a common grid? This would aid with reproducibility.

   **Response:** *Thank you for highlighting the need for reproducibility. In the ending statement of section 2.3.4 (formerly section 2.3.3) we mentioned that "For spatial representations, we interpolated the results of each model using the Climate Data Operator (CDO; Schulzweida, 2023) onto a uniform grid, using a spatial resolution already common to some of the ESMs: 0.94° x 1.25° (latitude x longitude). For extensive variables, such as land-use emission, we used conservative remapping with the `remapcon` function to preserve the integrals of the global totals (Jones, 1999). For intensive variables, such as temperature, we used bilinear interpolation with the `remapbil` function to preserve the mean values." We believe this clarifies the concern of the reviewer.*

5. Related to Table 2, is it possible that the average $\Delta$cLand is -131.9 (±96) GtC rather than -122 (±96) GtC?

   **Response:** *Thank you for noticing this. Yes, indeed. But in the caption of Table 2, an excerpt reads "[...] The model marked * (EC-Earth3-Veg*) is excluded from the multi-model mean of $\Delta$cLand because it has no fully activated carbon cycle." If this model was included, the multi-model mean of $\Delta$cLand would indeed be -131.9 (±96) GtC. But we excluded it from our computation due to the reason earlier given, resulting in a $\Delta$cLand value of -122 GtC.*

6. On page 14 where the methods for obtaining the grid cell temperature contribution and effect are discussed, is it possible to add a small amount of text to indicate the significance of or motivation for providing both quantities?

   **Response:** *Thank you for pointing this out. Following your suggestion, the new section 2.3.3. now reads, "We further attempt to distinguish between the grid cell temperature contribution and the grid cell temperature effect. The temperature contribution quantifies how much an individual grid cell's LUC adds to the global temperature signal, highlighting the locations that contribute most significantly to the global pattern. In contrast, the temperature effect measures how the climate in each specific location (grid-cell) is affected by global LUC, allowing us to assess localised impacts. Providing both quantities thus enables us to understand both the aggregate impact of LUC on global temperature and the specific local climate response to global LUC" The sentence in blue has been added to make our motivation more explicit.*

7. Page 25 line 511: "In magnitude, the warming pattern around Greenland can only be seen in the BGP contribution, which we attribute to mechanistic non-local LUC-induced effects on ocean currents and sea ice." This result seems worthy of mention in the discussion and conclusion sections.

   **Response:** *Thank you for highlighting this aspect of our research. We have now included this in the*

   *Discussion: "[…], the poleward warming contribution is due to the BGP effect alone, which includes both the local and non-local effects of LUC. For example, the warming pattern around Greenland seen only in the BGP contribution (Fig. 6c), can be attributed to mechanistic non-local LUC-induced effects on ocean currents and sea ice." and*

   *Conclusion: "[…] we find warming contributions over regions such as eastern Canada,*

*central Australia, and the tropics. We identified the warming contribution over the tropics resulting from the BGP effects as the only commonality between the BGP and BGC effects. In contrast, the warming pattern around Greenland can only be seen in the BGP contribution, which we attribute to mechanistic non-local LUC-induced effects on ocean currents and sea ice."*

8. Page 28 line 550: Is it possible to include the direction in which AMOC may have been influenced?

   **Response:** *We appreciate your interest in the potential influence of AMOC on the spatial patterns observed in our results. While we acknowledge that these patterns may suggest links to changes in simulated AMOC strength, a detailed analysis of AMOC behaviour is beyond the scope of our current study, which focuses primarily on the biogeophysical, and biogeochemical impacts of historical land-use change on near-surface air temperature and carbon dynamics.*

   *That said, we have included the statement […] namely Arctic and Antarctic sea-ice, and may have influenced the Atlantic meridional overturning circulation (AMOC) "The spatial temperature patterns in some models, particularly in higher latitudes, suggest links to AMOC changes. This interpretation aligns with findings in the broader literature, such as Weijer et al. (2020) (https://doi.org/10.1029/2019GL086075), which discusses AMOC behaviour in CMIP6 models, and other studies examining AMOC fingerprints (e.g., Rahmstorf 2024, https://doi.org/10.5670/oceanog.2024.501)". We suggest these references for further insights into AMOC-related changes in CMIP6 models and their implications for regional climate patterns.*

9. For the subplots in Figures 2-4 that represent just the direction of change and not the magnitude, is it possible to remove the numbers from the colorbars?

   **Response:** *Thank you for your suggestions. However, on closer review, we think the numbers below the colour bars help in identifying the number of models that agree on the direction of the signal. We have therefore modified the label to read "Number of Models" and added a statement in the figure caption that reads "[...] indicating the number of ESMs that agree on the direction of the signal."*

**Technical corrections**

1. Page 5 line 119: Please delete the "s" at the end of "backdrop."

   **Response:** *Right, we removed the "s".*

2. Page 13 line 313: The text reads "for a period ranging between 150 to 165." Please include the unit for "150 to 165" if units apply.

   **Response:** *Thank you for pointing this out. This has now been modified to "150 to 165 years"*

3. Page 17 line 403: In "- a trend" [...] "ΔcSoil -," please replace the hyphens with em-dashes.

   **Response:** *Thank you for spotting this. The hyphen has been replaced with em-dashes and we also checked the entire manuscript for similar occurrences.*

4. Page 24: The acronyms given in the caption for Figure 5 are not consistent with those given in the figure and in some places in the main text. Please adjust.

**Response:** *Thank you for pointing this out. The acronyms used in the figure caption have been modified to reflect those provided in the figure as well as in other places in the main text. For example, the Figure 5 caption now reads "The acronyms are NAT = North Atlantic, NAM = North America, EUR = Eurasia, SEB = South East Brazil, WAF = West Africa, SEA = Southeast Asia."*

5.  Page 27 line 543: In the sense that is likely intended, "widespread" ought to be written as "wide spread."

    **Response:** *Thank you for spotting this. The phrase has been changed from "widespread" to "wide spread"*

6.  Page 29: For Figure 7, is it possible to replace the dashed line separating the temperature (panels a and b) and carbon stock (panel c) with a solid line? This might further emphasize that data on two effects are presented in this figure.

    **Response:** *Thank you for your suggestion. The dashed line has been replaced with a simple line and the figure has been modified also in line with other comments received. The new Figure 7 is shown below*

[Figure]

**Figure 7.** Biogeophysical, biogeochemical effects, and changes in carbon stocks quantified in this study (hatched green bars) compared with other studies. Where vertical lines exist, they represent the standard deviation of estimates. See Supplementary Table S2 for the studies and their estimation periods.

7.  Page 30 line 609: Should "BGC and BGC" read "BGC and BGP"? If so, please change this.

    **Response:** *Yes, thank you for spotting this. This has been corrected and the statement now reads "[…] we show the aggregate of the BGC and BGP effects […]"*

8.  Page 36 line 791: Is it possible that temperature is being referred to here rather than "climate"?

    **Response:** *Thank you for bringing this to our attention. We revised this and the sentence now reads "In this study, we primarily focused on separating the temperature response caused by biogeophysical (BGP) effects of historical land-use change (LUC) from those caused by biogeochemical (BGC) effects. We go beyond previous studies to analyse the*

*most recent CMIP6 data, using state-of-the-art datasets contributed by the LUMIP project in an attempt to improve existing knowledge on the relative contribution of BGP and BGC effects of LUC on the "climate". However, we use the term "climate" here to collectively refer to effects beyond temperature alone.*

9. The supplementary information is very helpful for understanding the results model-by-model. It's possible that colorbars in figures S13, S14, S15, and S16 are a bit high compared to previous multi-model plots in the supplement

**Response:** *Indeed, the values in the colorbars of Figures S13 - S16 are high compared to the multi-model plots because they are in a different unit [%]. Here we show the percentage change across different land cover types*

**References**

*Jones, P. W. (1999). First- and Second-Order Conservative Remapping Schemes for Grids in Spherical Coordinates. Monthly Weather Review, 127(9), 2204–2210. https://doi.org/10.1175/1520-0493(1999)127<2204:FASOCR>2.0.CO;2*

*Rahmstorf, S. 2024. Is the Atlantic overturning circulation approaching a tipping point? Oceanography 37(3):16–29, https://doi.org/10.5670/oceanog.2024.501*

*Weijer, W., Cheng, W., Garuba, O.A., Hu, A., Nadiga, B.T. (2020) CMIP6 Models Predict Significant 21st Century Decline of the Atlantic Meridional Overturning Circulation. Geophysical Research Letters, 47(12): 47, https://doi.org/10.1029/2019GL086075*

---

## Author Comment (AC2)

**Response to Reviewer #2**

**General Comments**

This paper updates estimates of the biogeochemical and biogeophysical effects of land use change on surface temperature using CMIP6 output. They find large differences in estimated temperature effects across models, and generally similar magnitudes to prior work.

Overall, I find the main conclusions of the paper are somewhat buried. The text is very long, and the main take-home points are not very clear within the paper as a whole. There is extensive comparison with prior similar estimates and recapping of prior literature, but I didn't come away with an understanding for what was learned through this work specifically that wasn't already in previous work. This was especially the case in the discussion section. I understand that this paper uses different simulations than prior work, but what other insights the authors present that were not covered by previous literature I am unsure. I say this as someone slightly outside of the land use change sphere, where the detailed findings in the field are less known to me. I encourage the authors to make their specific contributions more clear in a revised version.

There are a lot of acronyms in this paper! (LUC, BGP, BGC, ESM, SNR, A/R, GHG, RGRT, TCRE, DGVM, all of the model names, etc). There is a lot to keep track of, especially for anyone not well embedded in this discipline. I suggest picking as few as you can that used extensively and spelling the rest out. For example, A/R, SNR, RGRT don't seem necessary as acronyms to me. Remove any you can reasonably remove to help out your readers!

**Response:** *Thank you for your feedback and the need to emphasise the novelty of our study. We have revised our discussion and conclusion sections to ensure the key messages are captured. At the beginning of each discussion section, we first highlight our novel contribution before expanding upon it in the following paragraphs. E.g.,*

*4.1. Disparity in estimates of near-surface air temperature across ESMs*

*"In highlighting the disparity across model estimates of near-surface temperature, we show that our findings align with previous studies in some aspects but also uncover critical deviations, particularly in the stronger BGC-induced warming observed in specific models. Unlike prior single-model studies or simplified model intercomparisons, we integrate multi-model analyses, spatial variability, and mechanistic insights into both regional and global BGP-BGC effects. Notably, we highlight how regional patterns—such as cooling over mid-latitudes and warming in the tropics—are shaped by complex interactions between BGP and BGC effects, including local and non-local feedbacks. While the global BGC-induced warming aligns with IPCC estimates and prior studies, the magnitude varies with LUC implementation details, such as gross vs. net transitions and forest cover representation. The BGP effects show greater inter-model disparity, largely influenced by differences in how vegetation fractions (e.g., tree cover) are modelled, affecting energy balance, albedo, and evapotranspiration. We expatiate on these findings below."*

*We repeat the same for subsections*

*4.2 Regional variability in BGP vs BGC effects on near-surface air temperature*

*"Results from our analysis in Sect. 3.3 confirm heterogeneous BGP effects, where LUC imprints on the temperature pattern, and homogeneous BGC effects. We provide summaries for more regions, with more models, than previous studies. Such regional information is important to anticipate how a region will be affected by LUC - important to know what to adapt to. In NAM for example - almost all models (more than in studies before, and with better LUC description and more processes) agree on the cooling, on average by 0.5 degrees. Given LUC needs to adapt to climate change, CDR needs and world economy, it is important to factor such benefits in to avoid bad surprises."*

and

*4.3. Variability across models' estimates of land-use change carbon emissions*

*"The variability in LUC-induced carbon emissions across ESMs reveals factors driving differences, including gross versus net transitions (Bayer et al., 2017; Bastos et al., 2022), initial carbon pool conditions (Boysen et al., 2021; Exbrayat et al., 2014), and model-specific treatments of factors like wood harvest (Hartung et al., 2021; Stocker et al., 2014) and irrigation (Qin et al., 2024; Roy et al., 2022). Our study builds on earlier work by quantifying the influence of these factors and examines their interactions in a multi-model framework using the latest ESMs complemented by a survey across modelling teams. Regional patterns confirm carbon losses in tropical regions due to deforestation (Zhu et al., 2023; Matricardi et al., 2020) and gains in Europe from land abandonment and regrowth (Ganzenmüller et al., 2022). Notably, gross transitions amplify flux estimates by capturing bidirectional land-use changes, while model-specific changes point to additional influences like irrigation and pre-industrial conditions (Melnikova et al., 2022). We therefore highlight the complexities of simulating LUC impacts and the critical need for harmonised modelling frameworks to improve the reliability and comparability of carbon flux projections across ESMs."*

To make the paper easier to read, we have removed the acronyms A/R, RGRT, and SNR

**Specific comments:**

1. Line 70: "local and non-local temperature changes" There is additional literature on this that is relevant, for example Laguë et al. 2019.

   **Response:** *Thank you for your suggestion. We have added the literature to the list of references.*

2. Line 309 - not clear how "RGRT" relates to the equations shown.

   **Response:** *Thank you for pointing this out. The RGRT, now written in full as the "regional-to-global ratio of temperature", at each grid cell, represents the ratio (a) of regional to global temperature change in that particular grid cell. It is used here to designate the rate of change of regional to global temperature, a model and grid cell-dependent metric.*

3. Line 331, equation7: I find this equation confusing to interpret. The authors acknowledge elsewhere in the manuscript that there are non-local effects of land cover change and that that this metric can't capture them because the temperature change in a single gridcell may be comprised already of both local and non-local impacts from land use change. Why then call it Tlocal? I'd suggest calling it something else that more accurately describes what it represents.

   **Response:** *Thank you for pointing this out. We have renamed the variable to better reflect that it also includes non-local effects for BGP. Equation 7 now reads as $T^{grid}$ which for both BGC and BGP. In the text, we explain that $T^{grid}$ for BGC comprises only the local effects, $T^{grid}$ for BGP comprises both the local and non-local effect.*

4. Line 358: "form a distinct cluster" - I do not see anything visually obvious like this in Figure 1. Needs further illustration or description.

   **Response:** *Thank you for pointing this out. We have now clarified our response to read* "In examining the trajectories of total land carbon change, $\Delta cLand$, we reveal considerable variation in how ESMs simulate changes in land $CO_2$ fluxes (Fig. 1a). This suggests that differences in $\Delta cLand$ trajectories are not directly relatable to differences in the implementation of LUC processes across models. Yet, the annual LUC emissions of CMCC, IPSL, and UKESM are very similar (Fig 1c), which might reflect that these models share a common approach by implementing net sub-grid transitions and simulating grasslands, but they do not represent pasture or grazing. For instance, sub-grid transitions allow models to account for mixed land-use types within a grid cell more precisely, leading to refined estimates of land carbon fluxes in areas where land use transitions over time. Moreover, focusing on grassland ecosystems rather than pasture or grazing may standardise the carbon flux response in these models, as grasslands generally have different carbon storage and release patterns than managed lands like pastures. Consequently, these shared characteristics could explain the observed alignment in land $CO_2$ flux trajectories by promoting a similar response to LUC across these models.*"

5. Line 360: "likely leads to similar trajectories" why? Please provide evidence and a hypothesis.

   **Response:** *We address these concerns in the earlier comment via a dedicated paragraph.*

6. Line 411: "across dynamic global vegetation models" do the authors mean TRENDY models? This is a confusing change of language, and additionally I don't think all TRENDY models are DGVMS - do the authors mean a subset of TRENDY?

   **Response:** *Thank you for pointing out the confusion that could arise from this. We follow the terminology used by the TRENDY team themself, which describes all the participating models as DGVMs (see Sitch et al., 2024, doi:10.1029/2024GB008102), irrespective of whether a dynamic biogeographical module is switched on or not (if this is what the reviewer alludes with their question of whether all the models are DGVMs). We have extended our text to clarify the terminology:*

*"[...] we compare [...] with estimates of dynamic global vegetation models (DGVMs) from the "Trends and drivers of the regional-scale sources and sinks of carbon dioxide" (TRENDY v11; Sitch et al., 2015) simulations"*

7. Line 450-451: see additional papers on how plant feedbacks with changing CO2 can amplify warming in high latitudes - Park et al. 2020 and Park et al., 2021. I think this literature is highly relevant to understanding the BGP effects of land use change.

   **Response:** *Thank you for recommending the papers by Park et al. (2020, 2021) regarding $CO_2$ physiological forcing and its implications for high-latitude warming. These studies provide valuable insights into the role of plant feedback in amplifying warming, particularly under $CO_2$ physiological forcing scenarios. However, our paper specifically addresses the effects of land-use change (LUC) forcing on biogeophysical (BGP) and biogeochemical (BGC) processes, which follow a different mechanism than the $CO_2$ physiological forcing examined in these studies. While we acknowledge that the role of plant feedback is an important and related area of research, the processes driving BGP and BGC effects under LUC are distinct from those associated with $CO_2$ physiological feedback. For this reason, we believe that a detailed discussion of $CO_2$ physiological forcing falls outside the primary scope of our analysis focused on LUC-driven effects.*

8. Figures 2-4: I suggest that the authors make the signal to noise ratio and inter-model agreement plots on a different color bar from the quantities being plotted (T, carbon) and the same colorbar for signal to noise ratio and inter-model agreement on all three plots. It is confusing to have it plotted in the same colorbar but showing a different unit.

   **Response:** *Thank you for your suggestions to improve clarity. We have modified Figures 2-4 as suggested, using a different colormap for the signal-to-noise ratio and the inter-model agreement. An example is shown below for Figure 2.*

[Figure]

**Figure 2.** Change in total land carbon pools (ΔcLand) as (**a**) the multi-model mean, (**b**) the inter-model spread, (**c**) the signal-to-noise ratio, and (**d**) the inter-model agreement due to biogeochemical effects of land-use change. Results were computed from 13 Earth system models as the cumulative value at the end of the simulation (year 2014). The signal-to-noise ratio (**c**) indicates the strength of the signal as compared to the inter-model uncertainty. It measures the relative weight of the multi-model mean anomalies in (**a**) with respect to the model coherence in (**b**) where a high absolute number means a robust signal. The inter-model agreement on the other hand shows the direction, rather than magnitude, of change for each grid cell (browns: negative/decreasing; greens: positive/increasing) indicating the number of ESMs that agree on the direction (+ or -) of the signal.

9. Line 496-497: "obviously driven foremost by the LUC in that gridcell" Can the authors say why this is obvious?

   **Response:** *Thank you for highlighting this. We revised the statement to provide clearer justification. The sentence now reads "While the underlying carbon stock changes in $\Delta T_{bgc}{}^{grid}$ are primarily driven by the LUC within the grid cell itself—since direct changes in land cover, vegetation type, and soil management directly affect carbon stocks at the local scale—, the resulting BGP temperature change in each grid cell reflects broader climatic impacts. These include changes in local surface properties (e.g., albedo, evapotranspiration) as well as energy and water vapour changes that may be caused by air transport into the grid cell originating from LUC in other locations."*

   *To justify these claims, we also added a statement in the methods, section 2.3.3., that reads "The underlying carbon stock changes in a grid cell are driven foremost by the LUC within that specific grid cell because our experimental setup isolates the effect of LUC by comparing two scenarios: historical and hist-noLu. By design, observed*

*differences in carbon stocks in a given grid cell are directly attributable to the local LUC imposed in that cell since this is the only variable altered between the two experiments. Therefore, the primary driver of carbon stock changes in each grid cell is the local LUC, as the experimental approach controls for other influences on carbon stocks."*

10. Line 501: "local BGP effects appear to dominate" How do the authors know that? I don't see evidence for this statement since they just stated that they can't calculate how much is local vs. non-local. Please provide further evidence.

    **Response:** *Thank you for pointing this out. We have clarified our statement by expanding on the text. The added clarification reads "The pattern of $\Delta T_{bgp}{}^{grid}$ is, therefore, a mixture of local and non-local effects of LUC (Winckler et al., 2019a), and the two effects cannot be separated without additional simulations. However, in regions with extensive LUC (see Figs. S13 - S16), such as areas experiencing substantial changes in vegetation cover or other land surface properties, it is reasonable to hypothesise that local BGP effects could have a more pronounced influence. Large-scale vegetation changes in these regions likely impact surface properties like albedo, evapotranspiration, and surface roughness, which are direct drivers of BGP effects. Thus, while our current approach cannot precisely quantify the local versus non-local contributions to $\Delta T_{bgp}{}^{grid}$ our maps provide an indication of areas where the unintended BGP effects of LUC are most likely significant. It is in this sense that our maps provide some guidance on the unintended effect of LUC in a specific location on global climate via BGP pathways—which again may be indicative of LUC deployed intentionally to dampen climate change; a consideration relevant for evaluating LUC as a strategy for climate mitigation."*

11. Figure 7. I think it would be helpful to label each of the three sections directly on the figure (biogeophysical, biogeochemical, change in carbon stock).

    **Response:** *In line with comments from other reviewers, we have modified Figure 7, and the new Figure 7 is as below.*

[Figure]

**Figure 7.** Biogeophysical, biogeochemical effects, and changes in carbon stocks quantified in this study (hatched green bars) compared with other studies. Where vertical lines exist, they represent the standard deviation of estimates. See Supplementary Table S2 for the studies and their estimation periods.

12. Line 686: "additional insights" - What about BCC and CESM2 provides insights? It isn't clear.

    **Response:** *Thank you for bringing this to our attention. We now provide further explanation to the insight reported in our manuscript. The statement reads "In examining regional variability in LUC effects, the U.S. Great Plains region offers insights into model differences, particularly in ΔcSoil estimates. While models such as BCC and CNRM indicate carbon gain in this region, CESM2 suggests carbon loss. This discrepancy highlights model-specific assumptions, including those related to grazing impacts on soil carbon. Derner et al. (2019) found that grazing does not significantly influence soil carbon levels in the Great Plains—a finding contested by Ren et al. (2024). Our analysis similarly finds no systematic relationship between ΔcSoil and models that implement grazing (see Table 1). This lack of alignment suggests that LUC effects on soil carbon in grazing systems may be highly model-dependent, underlining the complex interactions between LUC and regional soil carbon responses."*

13. Line 731-734: This explanation isn't clear to me. Also, what are the authors finding here about CESM2? Many of these paragraphs emphasize prior work but it isn't clear what is a new insight from this work.

    **Response:** *Thank you for highlighting the need for clarity. We have modified this to substantiate our claims and make our findings more pronounced. The refined statement reads thus: "We find that BCC and CESM2 are unique among the models in their responses to the gross versus net transition approach to LUC. BCC, which uses gross transitions, simulates carbon gain due to LUC, while CESM2, —which employs net transitions, —shows a high carbon loss, comparable in magnitude to models using gross transitions. This suggests that while the distinction between gross and net transitions generally explains model responses to LUC, it does not fully account for the behaviour observed in BCC and CESM2, indicating that additional processes likely influence these outcomes.*

    *Building on this observation, we note here that CESM2 stands out as one of the few models in our study (alongside EC-Earth3 models) that explicitly implements irrigation, a factor we find to correlate with high land CO2 flux estimates (Table 2). Our results suggest that irrigation could also be a contributing factor to the large carbon fluxes in CESM2, as irrigation has been shown in previous studies to increase LUC-related carbon fluxes (Qin et al., 2024; Roy et al., 2022; Taheripour et al., 2013) in addition to its BGP impacts on the land surface (De Hertog et al., 2023; Al-Yaari et al., 2022; de Vrese and Hagemann, 2018). According to de Vrese and Hagemann (2018), irrigation introduces heterogeneity within grid cells by increasing water availability in one part of the cell, creating sharp contrasts in land surface characteristics. This unique heterogeneity could help explain why CESM2's response to LUC differs from other models and why it shows high carbon loss despite using net transitions. Our findings thus highlight that model configurations like irrigation and the choice between gross versus net transitions interact in complex ways, affecting carbon flux outcomes in ways*

*not solely attributable to LUC representation."*

14. Line 745: "this underscores" I'm not sure what this is referring to? Is this a finding of this paper or of prior work?

    **Response:** *Thank you for highlighting the need for clarity. We have clarified the statement here and it now reads "According to Hartung et al. (2021), this contribution is larger than the uncertainty in LUC flux estimates, which affects cumulative the LUC flux by up to 22%. Together with these findings, our analysis highlights the necessity of including wood harvest in ESMs to achieve more accurate carbon flux estimations."*

---

## Author Response (AR2)

**We thank the two reviewers for taking their time to review our revised manuscript and for sharing valuable comments. Below, we provide pointwise response to comments raised where applicable.**

**Comments from Reviewer #1**

The authors have carried out a thorough and thoughtful revision that addresses all of my original comments in an appropriate and helpful way. The figure revisions and added points of clarification within the text are appreciated. While I agree with the other reviewer that the article is long and that there may be opportunities to make it somewhat more concise, there is value in the synthesis that documents differences in model configuration and how insights gained therein are applied to the interpretation of the study's results is likely to be quite helpful to others working in this specific domain of climate science. If shortening the article is deemed necessary to enhance appeal to a broader readership, I recommend that details removed from the main manuscript be transferred to the supplement so as to conserve the value such details add for those working in the same domain.

**Interdisciplinarity:**

Amali et al. 2024 is a work of interdisciplinary Earth system science in that the study of BGC and BGP temperature effects of historical LUC requires consideration of literature on LUC, land carbon sink, and atmospheric, sea ice, and ocean dynamics as well as how these components interact. This study therefore bridges several sub-disciplines of Earth system science. This study also considers feedbacks emerging from historical LUC in a non-trivial way.

**Scientific merits:**

As stated in the initial review:

"The study's findings both align with and expand upon previous work. For example, the finding that near-surface temperature increase from BGC is greater than BGP for historical LUC aligns with the findings within the existing literature. However, the regional analysis in Amali et al. adds nuance to this story in that the regional effect of BGP on near-surface temperature can be significant depending on location. Also significant is the study's contribution to our understanding of the BCG effect on near-surface temperature change at the gridcell level. Furthermore, the findings of this study demonstrates similar model spread and estimates to previous similar studies using LUCID or CMIP5 data, and identifies some reasons related to model architecture that contribute to this result."

The discussion is both thorough and balanced. Amali et al. add significant value through their consideration of individual model configurations and how these give way to the differences seen across the 13 models considered in this study. The authors neither understate nor overstate the significance of their conclusions.

**Technical quality:**

Are the scientific approach and applied methods valid? Are the results and conclusions presented in a clear, concise, and well structured way, including a reasonable number and quality of figures and tables and appropriate use of English language?

While the scientific approach and methods were initially considered to be sound, technical improvements to the text and figures have noticeably increased the manuscript's clarity. As mentioned above, it may be worthwhile to take the time to shorten this article to help it reach a wider audience, while conserving the significant value added in the presentation of differences in model configurations by moving these or other details less relevant to researchers in related fields to the supplement.

**Suitability:**

This manuscript certainly fulfils the criteria for an original research article. It is stylistically complete, containing all of the sections one would expect of such an article. In addition, all sections are thorough and well-developed.

**Recommendation to the editor:**

I recommend that this article be accepted subject to minor revisions to remove some of the detail from the main manuscript with the potential of relocating it to the supplement to enhance the article's appeal and useability for a broader audience within Earth system science.

**Response to Reviewer #1**

Dear Reviewer,

Thank you for your thoughtful and supportive feedback. We sincerely appreciate the time and effort you have taken to review our revisions and your recognition of the interdisciplinary value, scientific merit, and technical quality of our work.

We also appreciate your suggestion regarding shortening the manuscript to enhance accessibility for a broader audience. While we understand this concern, after careful consideration, we believe that the level of detail currently included is necessary to ensure the comprehensiveness and scientific rigor of the study. The synthesis of differences in model configurations and their implications for interpreting results is central to the study's contributions, and further condensation could risk omitting critical context needed for proper interpretation. Given that our findings aim to guide researchers working within this domain of Earth system science, we believe that maintaining the current structure best serves both clarity and completeness.

Once again, we truly appreciate your insightful comments and your support in strengthening our manuscript.

**Comments from Reviewer #2**

The authors have revised their initial submission and sufficiently addressed my previous comments. Below are some minor additional comments on this revised version.

1. line 374 - "For instance, sub-grid transitions allow models to account for mixed land-use types within a grid cell more precisely, leading to refined estimates of land carbon fluxes in areas where land use transitions over time." This sentence implies that the models not listed just before it do not have sub-grid transitions, but I believe that some of them do. Or maybe I don't understand the use of "sub-grid transitions" here. Could be clarified.

**Response:** *Thank you for bringing this to our attention. We have clarified the sentence, and it now reads: "For instance, annual LUC emissions of CanESM5, CMCC, IPSL, and UKESM are very similar (Fig. 1c), which might reflect that these models share a common approach: all of them implement net sub-grid transitions, explicitly consider explicit grassland simulations, and do not represent pasture or grazing. Models that implement net sub-grid transitions, such as these, allow for a more precise accounting of mixed land-use types within a grid cell, leading to refined estimates of land carbon fluxes in areas where land use transitions over time."*

2. 535 - "In magnitude, the warming pattern around Greenland can only be seen in the BGP contribution, which we attribute to mechanistic non-local LUC-induced effects on ocean currents and sea ice" (also a few other mentions of this warming around Greenland in the paper). This would be a relevant citation:
https://doi.org/10.5194/egusphere-2024-2087
Bauer, V., Schemm, S., Portmann, R., Zhang, J., Eirund, G. K., De Hertog, S. J., and Zibell, J.: Impacts of North American forest cover changes on the North Atlantic Ocean circulation, EGUsphere [preprint], https://doi.org/10.5194/egusphere-2024-2087

**Response:** *Thank you for your suggestion. We have added the citation to our manuscript.*

3. Figure 7. I think the labels help. If you put the bigoeophysical, biogeochemical, carbon labels at the bottom they would not interfere with the bar plots.

**Response:** *Thank you for your suggestion on Figure 7. We have modified the figure and placed the label at the bottom of the figure. The updated Figure 7 is as shown below.*

[Figure]

**Figure 7:** Biogeophysical, biogeochemical effects, and changes in carbon stocks quantified in this study (hatched green bars) compared with other studies. Where vertical lines exist, they represent the standard deviation of estimates. See Supplementary Table S2 for the studies and their estimation periods.